# A selectivity filter mutation provides insights into gating regulation of a K$^+$ channel

Theres Friesacher [1,5], Haritha P. Reddy [2,3,5], Harald Bernsteiner[1], J. Carlo Combista[2], Boris Shalomov[2], Amal K. Bera[3], Eva-Maria Zangerl-Plessl[1], Nathan Dascal [2,4 ✉] & Anna Stary-Weinzinger [1 ✉]

G-protein coupled inwardly rectifying potassium (GIRK) channels are key players in inhibitory neurotransmission in heart and brain. We conducted molecular dynamics simulations to investigate the effect of a selectivity filter (SF) mutation, G154S, on GIRK2 structure and function. We observe mutation-induced loss of selectivity, changes in ion occupancy and altered filter geometry. Unexpectedly, we reveal aberrant SF dynamics in the mutant to be correlated with motions in the binding site of the channel activator Gβγ. This coupling is corroborated by electrophysiological experiments, revealing that GIRK2$_{wt}$ activation by Gβγ reduces the affinity of Ba$^{2+}$ block. We further present a functional characterization of the human GIRK2$_{G154S}$ mutant validating our computational findings. This study identifies an allosteric connection between the SF and a crucial activator binding site. This allosteric gating mechanism may also apply to other potassium channels that are modulated by accessory proteins.

[1] Department of Pharmaceutical Sciences, Division of Pharmacology and Toxicology, University of Vienna, Josef-Holaubek-Platz 2, 1090 Vienna, Austria. [2] Department of Physiology and Pharmacology, School of Medicine, Tel Aviv University, Tel Aviv 69978, Israel. [3] Department of Biotechnology, Bhupat and Jyoti Mehta School of Biosciences, Indian Institute of Technology Madras, Chennai 600036, India. [4] Sagol School of Neuroscience, Tel Aviv University, Tel Aviv 69978, Israel. [5]These authors contributed equally: Theres Friesacher, Haritha P. Reddy. ✉email: dascaln@tauex.tau.ac.il; anna.stary@univie.ac.at

Potassium inward rectifier ($K_{ir}$) channels are key players in the regulation of the resting membrane potential. They constitute highly selective permeation pathways for $K^+$ ions across the cell membrane, which conduct inward currents more efficiently than outward currents. This phenomenon originates in a voltage-dependent block by intracellular $Mg^{2+}$ and polyamines preventing excessive $K^+$ efflux[1]. $K_{ir}$ channels form homo- as well as heterotetramers, which are generally activated by phosphatidylinositol 4,5-bisphosphate ($PIP_2$)[2]. Members of the $K_{ir}$ 3 (GIRK) channel subfamily are additionally regulated by intracellular sodium and the βγ subunits of the G-proteins, Gβγ (an obligatory dimer of Gβ and Gγ)[3]. The mechanism underlying the gating regulation of GIRKs by $Na^+$, $PIP_2$, and Gβγ is of great interest, since GIRKs have been shown to play crucial roles in a number of neurological processes, such as drug addiction, learning, and memory, as well as several neurological conditions, including Parkinson's disease, Down's syndrome, and epilepsy, turning them into crucial targets for drug development of neurological disorders[4].

In general, $K^+$ permeation through the GIRK channel pore is restricted by three distinct regions: the selectivity filter (SF) containing the highly conserved amino-acid motive TIGYG, the helix bundle crossing (HBC) gate, and the G-loop gate at the apex of the cytoplasmic domain (Fig. 1). So far, diverse aspects of GIRK channel control have been unraveled[5–12], but consensus on a gating model has not been reached yet. For example, X-ray crystallography experiments[5,6] suggest that HBC and G-loop gate opening is facilitated by $PIP_2$ and Gβγ in a co-dependent manner through a rotation of the C-terminal domain (CTD) and splaying of the inner helices. Interestingly, a crystal structure of GIRK2 bound to all of its activators does not capture a fully open channel, motivating the authors to propose a dynamic activated state, which fluctuates between non-conductive and conductive conformations[6]. Several MD simulation studies observe an influence of $PIP_2$ binding on the HBC gate[7,8]. Li et al.[9] hold Gβγ and $Na^+$ accountable for the opening of the G-loop gate and HBC gate, respectively, while reporting a stabilizing effect of $PIP_2$ binding on the open conformation of the gates. Another putative aspect of GIRK channel gating was revealed by Cryo-EM structures[10,11], which indicate that the CTD of the channel can adopt two different conformations: a docked and an extended

conformation that differ in the compactness as well as distance of the domain to the transmembrane part of the protein. $PIP_2$ binding leads to a switch from the extended to the docked conformation, making the channel susceptible towards Gβγ binding[10]. Such $PIP_2$-dependent conformational changes have also been reported for other $K_{ir}$ channels[12].

This study takes an approach towards investigating GIRK2 regulation by using a disease-causing mutation in the SF as a tool to look into allosteric connections governing channel activity. The mutation G154S in human GIRK2 (hGIRK2) is associated with a rare and severe neurological disorder, called the Keppen–Lubinsky syndrome (KPLBS, OMIM: 614098)[13]. While the biophysical properties of the disease mutant have not been studied yet, the corresponding mutation in mouse GIRK2 (mGIRK2) channels, the so-called *weaver* mouse model (mGIRK2$_{wv}$)[14–19], is known for more than 30 years. Studies detected remarkable changes in mGIRK2$_{wv}$ behavior, which is characterized by a loss of $K^+$ selectivity, insensitivity to blockage by extracellular $Ba^{2+}$ and elevated basal activity[14,16,17,19]. These severe abnormalities are linked to the death of cerebellar granule cells that underlies the disease symptoms of the *weaver* mouse[15,20].

Curiously, mGIRK2$_{wv}$ has previously been reported to display impaired activation by Gβγ[14,17,19]. This raises the question about the role of the SF region in controlling GIRK2 activation by physiological stimuli. Despite its pathological importance, experimental characterization of the biophysical, physiological, and pharmacological properties of hGIRK2$_{G154S}$ is still lacking.

In this work, we use μs-long MD simulations to investigate the effect of the *weaver* mutation on mGIRK2, which we corroborate by a functional characterization of the corresponding human mutant, hGIRK2$_{G154S}$, with electrophysiology experiments. We provide insight into the functional, dynamic, and allosteric consequences of the SF mutation. We reveal that the mutation causes abnormalities in channel behavior and unveil SF alterations that provide mechanistic understanding of the loss of $K^+$ selectivity. By subsequently applying a functional mode analysis (FMA), we identify collective motions, which are coupled to aberrant SF dynamics in the mutant channel. The analysis uncovered an unexpected correlation between SF movements and dynamics in the Gβγ-binding site. This finding was corroborated by electrophysiological experiments that reveal an effect of Gβγ on hGIRK2$_{wt}$ block by the pore blocker $Ba^{2+}$. Our study uncovers a novel aspect of GIRK2 gating, which is characterized by a coupling of the of the SF to a major regulator binding site.

## Results

**The *weaver* mutation leads to a loss of selectivity and efficiency of $K^+$ permeation.** We conducted μs-long MD simulations of the wild-type (mGIRK2$_{wt}$) and mutant (mGIRK2$_{wv}$) channel in order to investigate the consequences of the weaver mutation on channel dynamics. From here on, the mouse *weaver* mutant mGIRK2$_{G156S}$ is termed mGIRK2$_{wv}$ in order to avoid confusion with the hGIRK2 mutant G154S, here named hGIRK2$_{G154S}$. An open and conductive state from our previous simulations of the $PIP_2$-bound mGIRK2$_{wt}$ channel was extracted[7] (PDB: 3SYA). The channels were simulated in mixed $Na^+/K^+$ salt conditions (90 mM KCl:90 mM NaCl) applying electric fields in the range of 40–60 mV nm$^{-1}$ in the outward direction. Supplementary Table 1 summarizes all runs and the observed ion permeation events. A total of 6 μs simulation data with mGIRK2$_{wt}$ unraveled a high selectivity for $K^+$, yielding a permeability ratio $p_{Na}/p_K$ of 0.025 ($p_K/p_{Na} = 40$). In contrast, 9 μs simulation data of the mGIRK2$_{wv}$ channel revealed an increase of $p_{Na}/p_K$ to 0.31 ($p_K/p_{Na} = 3.22$) (Supplementary Table 1). Our results concur with a previous electrophysiological study that reports a $P_K/P_{Na}$ ratio

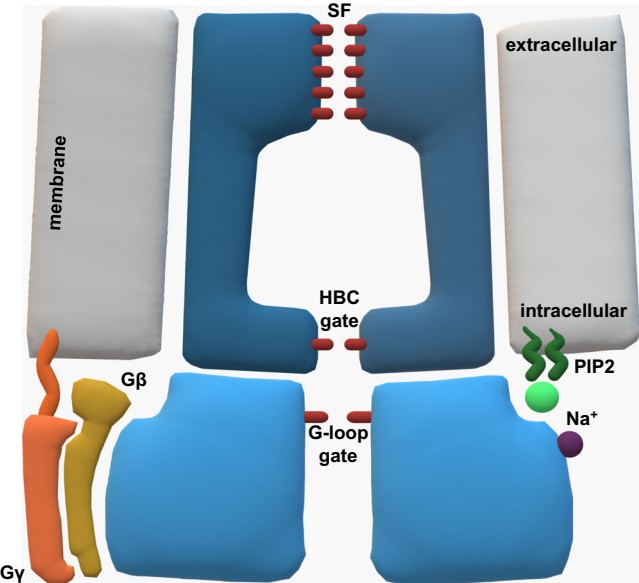

**Fig. 1 The structure of the GIRK2 channel and its regulators.** Two of the four subunits of the channel as well as channel modulators are illustrated.

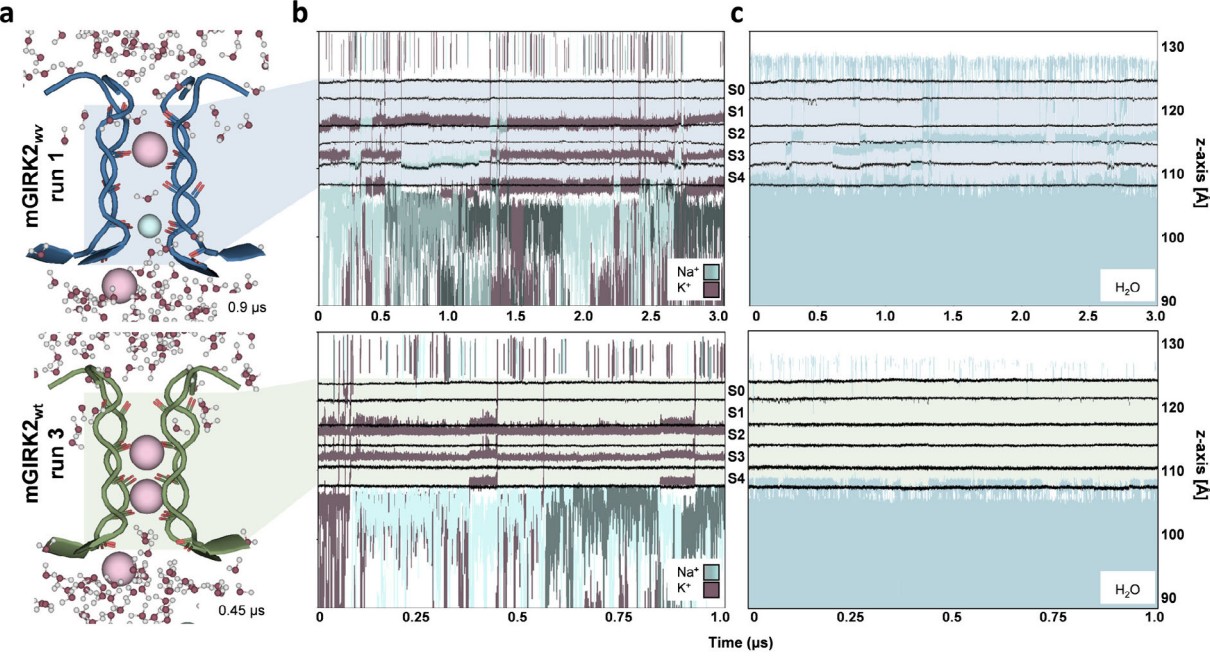

**Fig. 2 Ion and water flux through the mGIRK2$_{wv}$ (top) and mGIRK2$_{wt}$ (bottom) SF over simulation time. a** Representative snapshots of ion and water flux through the SF. mGIRK2$_{wv}$ is colored blue, while mGIRK2$_{wt}$ is colored green. The SF is depicted in cartoon representation with the backbone oxygens of the SF residues indicated as red sticks. K$^+$ ions are represented as pink spheres, while Na$^+$ ions are represented as smaller, light blue spheres. Water molecules are shown in a ball-and-stick representation. **b** K$^+$ (purple) and Na$^+$ ions (blue) flux observed in runs with mGIRK2$_{wv}$ and mGIRK2$_{wt}$. Each line displays the position of one ion on the $z$-axis over simulation time. The positions of the backbone oxygens of the SF residues, which frame the canonical ion binding sites S0–S4, are shown in black. **c** Presence of water in the mGIRK2$_{wv}$ and mGIRK2$_{wt}$. Each line displays the position of one water molecule on the $z$-axis over simulation time. The positions of the backbone oxygens of the SF residues are shown in black.

of ≥20 for the mGIRK2$_{wt}$, which is contrasted by $P_K/P_{Na}$ ratios of ~2 for mGIRK2$_{wv}$[19].

Figure 2b shows the outward permeation of ions and the presence of water in the mGIRK2$_{wv}$ SF and mGIRK2$_{wt}$ SF over simulation time. The most striking divergence between the simulations is the mutation-induced loss of selectivity for K$^+$, which is demonstrated by several Na$^+$ passing through the mGIRK2$_{wv}$ SF. The relative Na$^+$ permeation rate increases from 0.16 ions/µs for mGIRK2$_{wt}$ to 2.25 ions/µs for mGIRK2$_{wv}$. Furthermore, our simulations unveil reduced K$^+$ movement through the mutant channels, which is reflected in the drop of relative K$^+$ permeation rates from 6.67 ions/µs for mGIRK2$_{wt}$ to 1.4 ions/µs for mGIRK2$_{wv}$.

Interestingly, mGIRK2 conductance varies greatly between the different runs. In case of mGIRK2$_{wv}$, one of the six runs is accountable for 10 of 13 conduction events, while in four runs no ions permeated. Similarly, run 3 of the mGIRK2$_{wt}$ shows extreme conductance. Hence, mGIRK2 is not steadily permeable in our simulations, but transiently adopts a conductive state allowing multiple ions to pass within a short period of time (Fig. 2 and Supplementary Fig. 1).

mGIRK2$_{wt}$ block by Na$^+$ was observed in three of the six wild-type runs. In mGIRK2$_{wt}$ run 1 and run 6, ion flux is halted shortly after simulation start due to the binding of a Na$^+$ ion between the sites S3 and S4 (Supplementary Fig. 1). In run 5, Na$^+$ advances to site S2 and remains there stably bound, thereby inhibiting further ion permeation. The block of K$^+$ flux by Na$^+$ ions is in agreement with previous electrophysiology experiments and simulations studies[19,21–23].

In the simulations with mGIRK2$_{wv}$, it can be observed that raised Na$^+$ permeation levels are accompanied by a deficient

exclusion of water from the SF. The right permeation plot in Fig. 2b shows water entering the mutant SF after about 250 ns simulation time, occurring simultaneously to the first Na$^+$ permeation event. However, solvation is not tightly coupled to Na$^+$ permeation, since water is also present in case only K$^+$ occupies the SF (e.g., Fig. 2b between 1.5 and 2 µs). On the contrary and in line with previous studies[7,21,24], no water can be seen in the simulations with mGIRK2$_{wt}$. K$^+$ ions shed the water shell upon entering the SF and permeate in a dehydrated manner.

**Experimental characterization of the hGIRK2$_{G154S}$ reveals drastic changes in channel selectivity and regulation.** We used two-electrode voltage-clamp experiments to carry out the functional characterization of hGIRK2$_{G154S}$ with the aim to investigate the effects of the SF mutation on human GIRK2 behavior. hGIRK2$_{wt}$ and hGIRK2$_{G154S}$ channels were expressed in Xenopus oocytes by injecting the corresponding RNAs. hGIRK2 activation was achieved by coexpression of Gβγ[25,26]. We compared current amplitudes and current–voltage (I–V) relationships in 5 groups of oocytes either by expressing hGIRK2$_{wt}$ with or without Gβγ, or hGIRK2$_{G154S}$ with or without Gβγ, or naive oocytes (uninjected with any RNA).

Figure 3a shows a representative recording from oocytes expressing hGIRK2$_{wt}$ with Gβγ. The oocyte was sequentially exposed to four extracellular solutions containing 2, 8, 24, or 72 mM K$^+$ and, correspondingly, 96, 90, 74, or 26 mM Na$^+$. A voltage ramp from −120 to 50 mV was applied in each solution (Fig. 3b), yielding I–V curves (Fig. 3c). The same procedure was repeated in the four solutions containing 1 mM Ba$^{2+}$, which is a pore blocker of K$_{ir}$ channels[23,27–29]. To allow comparison of

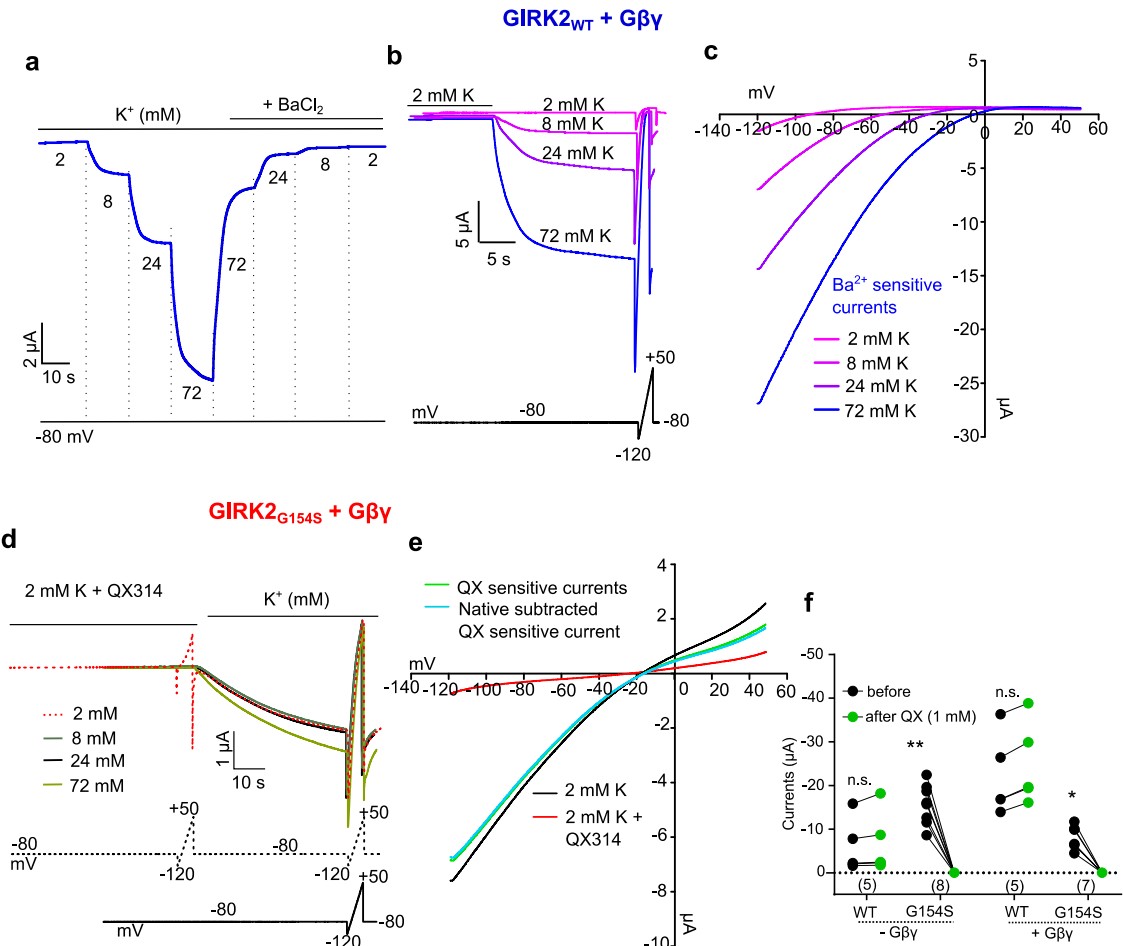

**Fig. 3 hGIRK2_{G154S} compared to hGIRK2_{wt}.** **a** Representative of whole cell current records from hGIRK2_{wt} with Gβγ. Sequential changes of solutions and subsequent Ba²⁺ block are shown. The holding potential was −80 mV, and the recording always started in low-K⁺ (2 mM) solution. **b** Representative recordings of whole-cell currents from hGIRK2_{wt} with Gβγ. 2 s voltage ramps from −120 to +50 mV are performed in each solution. Record from each solution is shown in a different color. Voltage protocol is shown in the lower panel. **c** I–V curves in a representative oocyte expressing hGIRK2_{wt} with Gβγ tested in four solutions. Ba²⁺ sensitive currents are shown. **d** Representative recordings of whole-cell currents from hGIRK2_{G154S} with Gβγ. 2 s voltage ramps from −120 mV to +50 mV are performed in each solution. **e** I-V curves in a representative oocyte expressing hGIRK2_{G154S} with G_{βγ} in the 2 mM K⁺ solution (black) along with QX block (red), QX sensitive currents (green) and the net QX currents obtained from subtracting average native currents from QX sensitive currents (cyan) are shown. **f** hGIRK2_{wt} and hGIRK2_{G154S} currents with and without Gβγ, before and after QX314 (1 mM) application, in 4–7 oocytes from one frog, are shown. Paired *t*-test followed by Wilcoxon test was performed. **p = 0.0078 and *p = 0.0156; n.s. is not significant (p > 0.05).

current amplitudes between the different experimental groups and with previous publications[30,31], we used the values of inward currents at −80 mV in 24 mM K⁺ solution (Fig. 4a).

In line with previous studies[30], uninjected oocytes showed small, mostly leak currents of −51 ± 4 nA (n = 5, Supplementary Fig. 2a), and only a minor fraction (2 ± 1 nA, n = 5) was blocked by Ba²⁺. In contrast, currents in hGIRK2_{wt} and hGIRK2_{wt} + Gβγ groups were almost fully blocked by Ba²⁺, therefore net hGIRK2_{wt} I–V curves were obtained by subtracting Ba²⁺-insensitive currents[26] (Fig. 3c and Supplementary Fig. 2b–d). As expected, hGIRK2_{wt} showed inward currents and strong inward rectification both in the presence (Fig. 3c) and absence (Supplementary Fig. 2c) of Gβγ, which were increased in amplitude with increased [K_o] (Figs. 3b and 4b). In contrast, hGIRK2_{G154S} yielded large basal currents (Figs. 3d, e, 4a, b and Supplementary Fig. 2e–g), similar to its mouse counterpart, the mGIRK2_{wv}[16]. The amplitude of hGIRK2_{G154S} currents, both with or without Gβγ, was >80 times greater than in native oocytes (−4.2 ± 0.4 μA (n = 6) and −6.5 ± 1.5 μA (n = 6), respectively) (Fig. 4a). Therefore, for further analysis, we assumed that the total current measured in these cells represents the net hGIRK2_{G154S}

current with >95% accuracy. I–V curves of hGIRK2_{G154S} with or without Gβγ showed little rectification (Fig. 4f) and reduced block by Ba²⁺ (~60% block at 1 mM Ba²⁺, compared to full block of hGIRK2_{wt}) (Supplementary Fig. 2g). Similar behavior was previously reported for the mouse *weaver* mutant, which was also observed to be blocked by the cation channel blocker QX314[14,19]. Accordingly, GIRK2_{G154S} was blocked by 99.6 ± 0.1% by 1 mM QX314, while hGIRK2_{wt} was unaffected by QX314 (Fig. 3f).

Figure 4 shows the quantitative analysis of whole-cell hGIRK2_{wt} and hGIRK2_{G154S} currents. As previously reported for mGIRK2_{wt}[30], hGIRK2_{wt} yielded small I_{basal}, −194 ± 49 nA (n = 5) and showed strong, >18 fold activation by coexpressed Gβγ (−3.6 ± 1.7 μA, n = 6). In contrast, hGIRK2_{G254S} yielded a very large I_{basal}, which was not significantly increased by the coexpression of Gβγ (Fig. 4a). As noted above, hGIRK2_{wt} currents rised in amplitude with increased [K_o] (K⁺ was replacing Na⁺ in the external solution), whereas hGIRK2_{G154S} currents were almost insensitive to [K_o] (Fig. 4b), suggesting that K⁺ was not the sole carrier of the inward current at −80 mV. Figure 4c shows the dependence of the reversal potential (V_{rev}) of

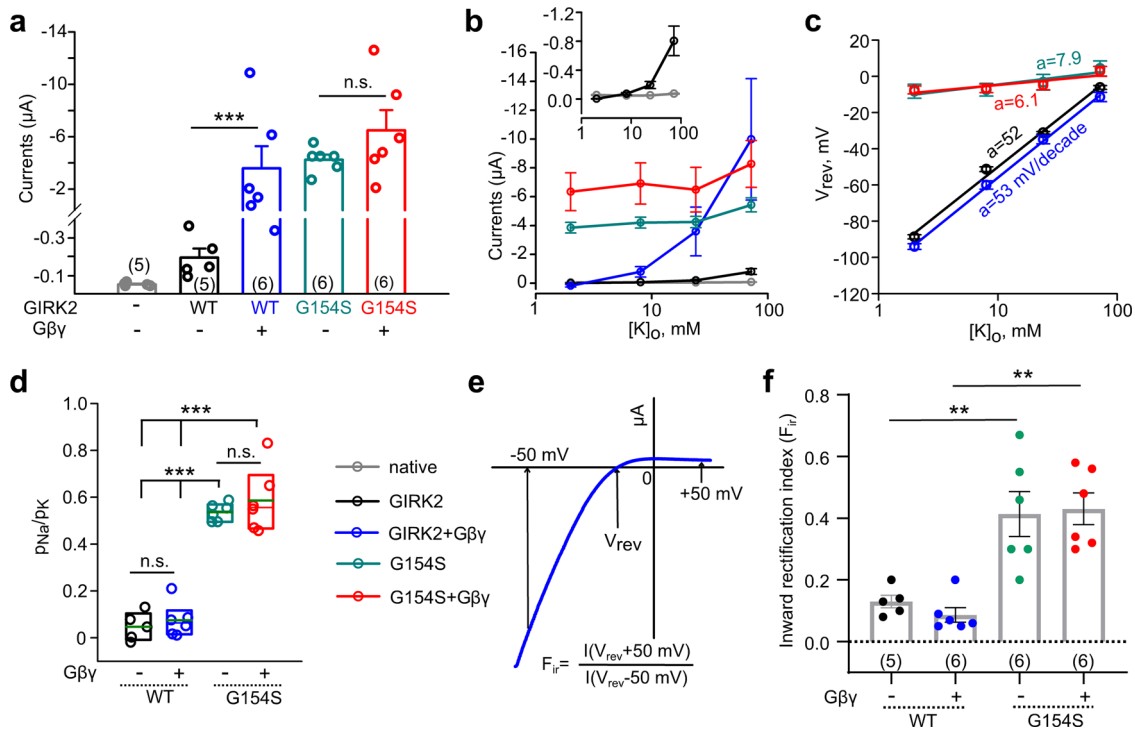

**Fig. 4 hGIRK2$_{G154S}$ channels are constitutively active and lose selectivity to K$^+$. a** Comparison of current amplitudes at −80 mV in 24 K$^+$. Gβγ activates hGIRK2$_{wt}$ (unpaired two-tailed $t$-test) but not hGIRK2$_{G154S}$. $n = 5$–7 oocytes (from one frog) in each group. The data in the bar graphs, here and in **f**, are presented as mean ± SEM. **b** Changes in current amplitudes as a function of external K$^+$ (note that K$^+$ isotonically replaced Na$^+$, Cl$^−$ was kept constant). The inset zooms on native and hGIRK2$_{wt}$ groups that showed lower currents. **c** The dependence of $V_{rev}$ on [K$_o$]. Note the log scale of the $X$-axis. The slopes (**a**; mV/decade) of the linear least square regression lines are indicated near each line. Data are shown as mean ± SEM from 6 to 7 oocytes in each group, from one frog. **d** $p_{Na}/p_K$ permeability ratios, calculated from the reversal potentials obtained in 8 and 72 mM K$^+$ solutions, in the four test groups. Kruskal–Wallis one-way ANOVA on ranks, followed by Dunn's pairwise multiple comparison test. In box plots, boxes show 25th and 75th percentiles. The horizontal lines colored as the box show the median, the green line shows the mean. ***$p < 0.001$. **e** An example of calculation of the rectification index, F$_{ir}$. **f** Comparison of F$_{ir}$ of hGIRK2$_{wt}$ and hGIRK2$_{G154S}$ channels with and without Gβγ. **$p = 0.0065$ and 0.0022. Numbers of oocytes tested are indicated below the bars.

hGIRK2$_{wt}$ and hGIRK2$_{G154S}$ on [K$_o$] (note the log scale on the $X$-axis). For hGIRK2$_{wt}$, $V_{rev}$ was linearly related to log[K$_o$], with a slope of the $V_{rev}$−log[K$_o$] curve of 52–53 mV per 10-fold change in [K$_o$] (decade), close to the predicted slope of 58 mV/decade for a perfect K$^+$ channel. Inversely, for hGIRK2$_{G154S}$, the change in $V_{rev}$ was rather small, with a slope of 6–8 mV/decade, supporting lack of selectivity for K$^+$. We subsequently calculated the permeability ratios for K$^+$ to Na$^+$, $r = p_{Na}/p_K$, by comparing the values of $V_{rev}$ in two solutions containing different concentrations of Na$^+$ and K$^+$. The calculated values of $p_{Na}/p_K$ for the 8–72 mM K$^+$ pair are shown in Fig. 4d. In presence of Gβγ, $p_{Na}/p_K$ ratios of $0.029 \pm 0.004$ ($n = 6$) for hGIRK2$_{wt}$ and $0.59 \pm 0.05$ ($n = 7$) for the hGIRK2$_{G154S}$ were calculated. In case Gβγ was not co-expressed, we assessed $p_{Na}/p_K$ of $0.05 \pm 0.01$ ($n = 5$) and $0.52 \pm 0.04$ ($n = 6$) for hGIRK2$_{wt}$ and hGIRK2$_{G154S}$, respectively. Importantly, these ratios are in good agreement with our Gβγ-free simulations, which yielded $p_{Na}/p_K$ ratios of 0.025 and 0.31 for hGIRK2$_{wt}$ and hGIRK2$_{G154S}$, respectively. Additionally, the 20–30 fold higher $p_K$ relative to $p_{Na}$ for hGIRK2$_{wt}$ is in good agreement with a previous report[19].

We also quantified the extent of inward rectification (F$_{ir}$) (Fig. 4e, f). F$_{ir}$ of hGIRK2$_{wt}$ channels in presence and absence of Gβγ was $0.08 \pm 0.009$ ($n = 6$) and $0.13 \pm 0.009$ ($n = 5$), respectively, indicating strong inward rectification. For hGIRK2$_{G154S}$, we calculated F$_{ir}$ values of $0.41 \pm 0.02$ ($n = 6$) with Gβγ and $0.43 \pm 0.02$ ($n = 6$) without Gβγ, indicating a loss of rectification.

**Ion occupancies in the mutant SF are shifted towards the extracellular site.** Since an important aim of this framework is to elucidate not only the functional consequences of the SF mutation, but also the mechanism behind the channel malfunction, we undertook further analysis of our MD simulations focusing on the impact of the mutation at an atomistic scale. In a first step, we compared the ion occupancies in the SF, which unveiled clear differences between the wild-type and the mutant channel (Fig. 5a). In the mGIRK2$_{wt}$ SF, K$^+$ mainly populates the binding sites S2 and S3. The latter is also occupied in mGIRK2$_{wv}$, while the peak at S2 is shifted towards the extracellular site, so that K$^+$ binds at an in-plane binding-pose between the canonical ion binding sites S1 and S2, at the level of the mutated residue S156. Interestingly, the same site is extensively occupied by Na$^+$ in mGIRK2$_{wt}$, which, together with another in-plane conformation between sites S3 and S4, originates in the blockage of mGIRK2 by Na$^+$ ions. The occupation of positions between classical binding sites by non-K$^+$ ions has also been described in previous studies[21,32–35]. On the contrary, negligible Na$^+$ occupancies in the mGIRK2$_{wv}$ SF can be observed, which indicates lower energy barriers for Na$^+$ permeation and are interpretable as the basis for enhanced Na$^+$ permeation rates.

**S156 can form hydrogen bonds with the backbone oxygen of I155 and water molecules behind the SF.** A closer investigation of the SF revealed that the mutant channel gains the ability to form a hydrogen bond between the sidechain of the mutated

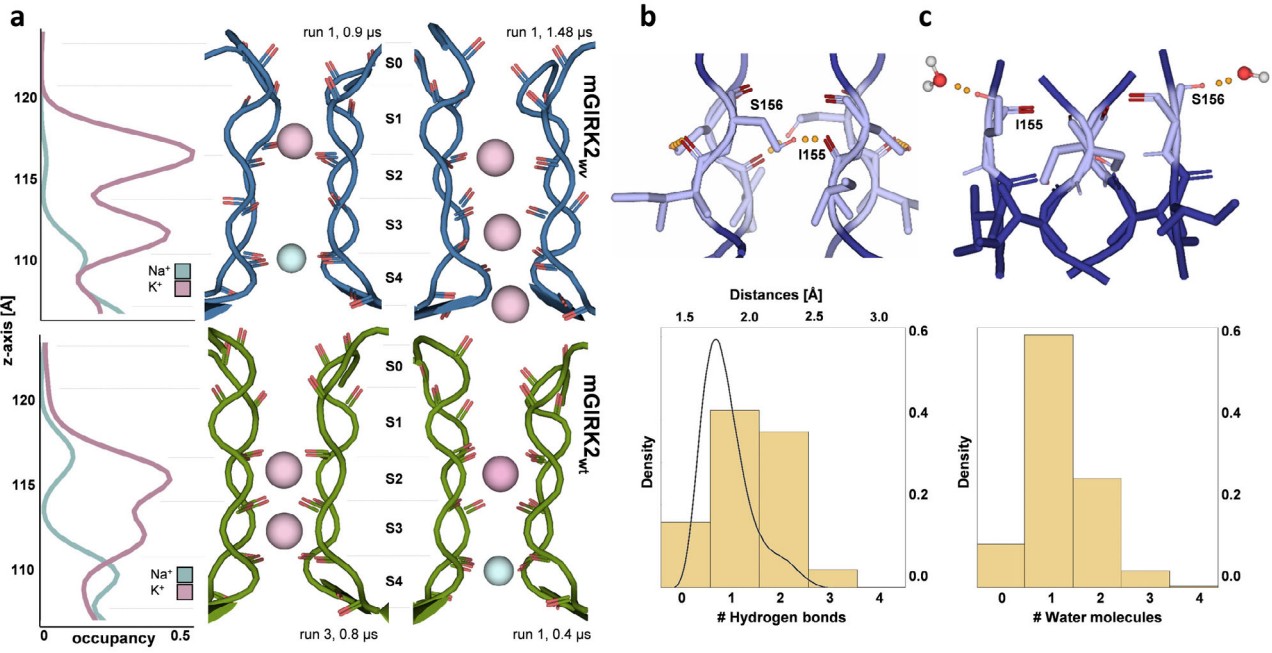

**Fig. 5 Differences in ion occupancies and in the hydrogen bond network between mGIRK2$_{wt}$ and mGIRK2$_{wv}$. a** Top: Ion occupancies in the mGIRK2$_{wv}$ SF observed over a total of 9 μs-long MD simulation. Two characteristic snapshots of Na$^+$ and K$^+$ permeation are shown on the right. Bottom: Ion occupancies in the wild-type SF observed over 6 μs long MD simulations. The snapshots on the right show a characteristic K$^+$ occupancy pattern as well as Na$^+$ block between sites S3 and S4. **b** Top: Representative snapshot of hydrogen bonds between the sidechain of S156 and the backbone oxygen of I155 in the mGIRK2$_{wv}$ SF. Bottom: Density distribution of the distances and the numbers of hydrogen bonds between the sidechain of S156 and the backbone oxygen of I155 observed over 9 μs simulations with GIRK2$_{wv}$. **c** Top: Representative snapshot of hydrogen bonds between the sidechain of S156 and a water behind the mGIRK2$_{wv}$ SF. Bottom: Histogram of the number of water molecules behind the SF, which form hydrogen bonds to the sidechain of S156, over 9 μs simulations with mGIRK2$_{wv}$.

residue S156 and the backbone oxygen of I155 (Fig. 5b). Looking at the occurrence of this interaction in the simulations with mGIRK2$_{wv}$, it can be seen that the hydrogen bond exists between one or two subunits for a substantial fraction of the simulation frames. The plot at the bottom of Fig. 5b shows a histogram of the number of hydrogens bonds as well as the distribution of the distances between S156 and the backbone oxygen of the I155. The latter displays an accumulation of the distances between 1.5 and 2 Å, further confirming the importance of this interaction. Moreover, the sidechains of S156 were observed to form hydrogen bonds with water molecules, which occasionally diffuse behind the SF (Fig. 5c). Subsequent investigations revealed that this kind of interaction is most commonly established by one of the four subunits during the simulation. The mGIRK2$_{wt}$ SF harbors a glycine at this site and hence lacks these functionalities.

**The HBC gate is not affected by the SF mutation**. In order to analyze whether the HBC gate is influenced by the SF mutation, we measured the distances between F192 of opposing subunits. For both mGIRK2$_{wt}$ and mGIRK2$_{wv}$, the distribution of distances between opposing F192 Cα atoms culminates at about 18.5 Å, while the peak of the minimum distance between opposing F192 residues is at around 11 Å (Supplementary Fig. 23). In order to identify the motions in the channel maximally correlated with the movements of the HBC gate, Functional Mode Analyses (FMAs) were conducted. FMA models were constructed using the minimum distances between opposing F192 residues and the distances between F192 Cα atoms as functional quantities. None of the models obtained for mGIRK2$_{wv}$ or mGIRK2$_{wt}$ exceeded correlation measures of 0.23 in cross-validation. Thus, the MD

simulations show that the HBC gate is open to a similar degree in mutant and wild-type mGIRK2 and its movements are not correlated to any other motions in the ion channel. However, it should be emphasized that an open state of mGIRK2 was used as starting point of the simulations[7]. A coupling of the SF movement as well as an impact of the mutation on a more closed HBC gate can therefore not be excluded and remains to be elucidated in future studies.

**Mutation-induced structural differences in the SF are associated with movements in the Gβγ-binding site**. In order to obtain detailed insight into the structural mechanisms underlying the alterations in ion selectivity and occupancies, we analyzed the geometry of the wild-type and mutant SF in detail. For this purpose, φ angles as well as inter-subunit distances of Cα atoms of the SF composing residues 154–158 were monitored over time (Fig. 6b). Analysis of the φ angle distribution revealed a striking difference at the mutated residue S156, displaying a pronounced peak at +60°, which is also sampled in mGIRK2$_{wt}$ for G156, as well as a peak at −60°. It is likely that this new conformation is a side effect of the hydrogen bond between the S156 sidechain and the backbone oxygen of I155, which was identified as a quality of the mutant SF (Fig. 5b). Apart from the S156, reorientation of the carbonyl oxygen atoms away from the conduction pore is rarely observed. Some conformational flexibility can be seen at the extracellular entrance of the SF, as well. However, this plasticity at the threonine 154 occurs independently of the SF mutation in both mGIRK2$_{wt}$ and mGIRK2$_{wv}$.

When looking at the distances between the Cα atoms of the SF-lining residues, it is noticeable that the mutant exhibits a wider

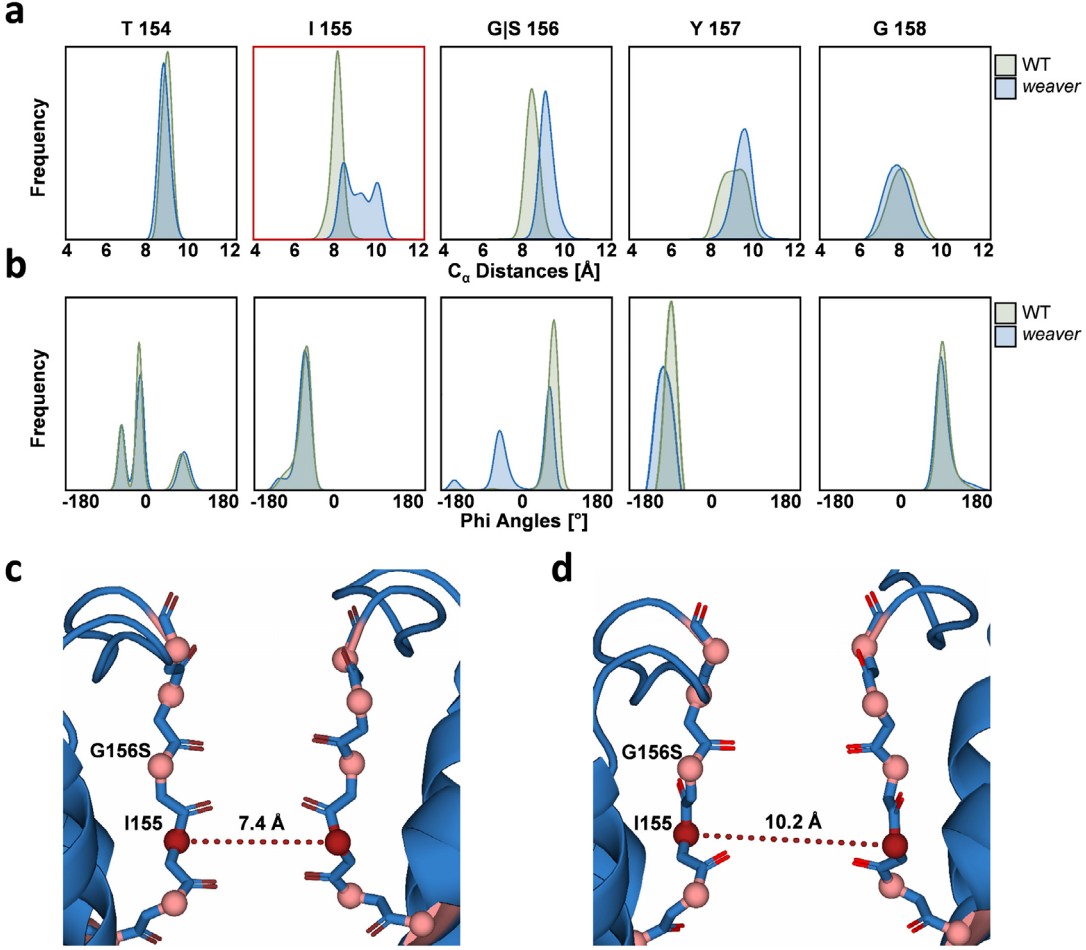

**Fig. 6 Structural aberrations in the SF observed over 9 µs mGIRK2$_{wv}$ and 6 µs mGIRK2$_{wt}$ simulation. a** Distributions of distances between Cα of opposing mGIRK2 SF residues. **b** Phi-angle distributions of mGIRK2 SF residues. **c** and **d** mGIRK2$_{wv}$ SF conformations showing the minimum (**c**) and maximum (**d**) distance between opposing I155 Cα atoms. The Cα atoms of I155 is highlighted are shown in dark red. Other Cα atoms of the SF residues are shown as salmon spheres.

conformation at the residues 155–157, which confine the ion binding sites S2 and S3 within the SF filter (Fig. 6a). This difference is most pronounced for the isoleucine 155, which displays a bimodal distribution of inter-subunit distances with the two extremes about 2 Å apart (Fig. 6c). Considering the importance of SF geometry of K$^+$ channels for selectivity and activity[32,36], this change in the SF conformation is drastic and is likely to provoke profound alterations in channel behavior. In our simulations, the dilation might play a profound role in the decrease in K$^+$ selectivity and an enhanced accessibility of the SF for water molecules.

In order to elucidate the dynamic consequences of the aberrant motions in the mGIRK2$_{wv}$ SF, we used the functional mode analysis (FMA) tool[37] to identify the collective motion maximally correlated with I155 fluctuation. For this purpose, movements of the protein backbone were analyzed. The highly dynamic extracellular loop (residues 120–133) and C-terminus (residue 369–382) were excluded in order to reduce the background noise. Remarkably, the FMA identified mainly movements in the C-terminal domain (CTD) of mGIRK2$_{wv}$ to be linearly correlated with the I155 distance fluctuations (Fig. 7). Cross-validation of the model yielded a Pearson $R$ of 0.73 (Fig. 7d).

When taking a closer look at the correlated movements in the CTD, it can be seen that the region determined to be most affected by I155 distance fluctuations comprises the residues

247–254. Importantly, these amino acids constitute the βD-βE loop of the GIRK2 CTD, which, together with the βL-βM loop of the adjacent subunit, composes the Gβγ-binding site in the Gβγ-mGIRK2 crystal structure[6] (Fig. 7b). The SF and the Gβγ-binding site are located at the upper region of the transmembrane domain and at the inferior side of the CTD, respectively, resulting in a distance of about 75 Å. In spite of the spatial segregation, the regions separating the SF and the Gβγ-binding site seem to be largely unaffected by the dynamics of I155 indicating that the correlation is not the result of a mutation-induced global rearrangement, but an allosteric coupling between the mutant SF and a crucial regulatory site.

The FMA identified further correlated movements at the lowest part of the inner loops of the CTD around the residue Y267 (Fig. 7b). An involvement of this region in channel regulation is not supported by the literature. Nevertheless, a regulatory function cannot be entirely excluded due to its crucial location at the intracellular entry of the GIRK2 pore. Moreover, less pronounced coupling could be seen in the lower part of the transmembrane helix 1 and the region below the tether helix (Fig. 7c).

An FMA of I155 dynamics in mGIRK2$_{wt}$ could not unveil significant correlations with any part of the channel (Fig. 7d, Pearson $R \sim 0.35$). A possible explanation for the absence of correlation might be that the coupling between the SF and the

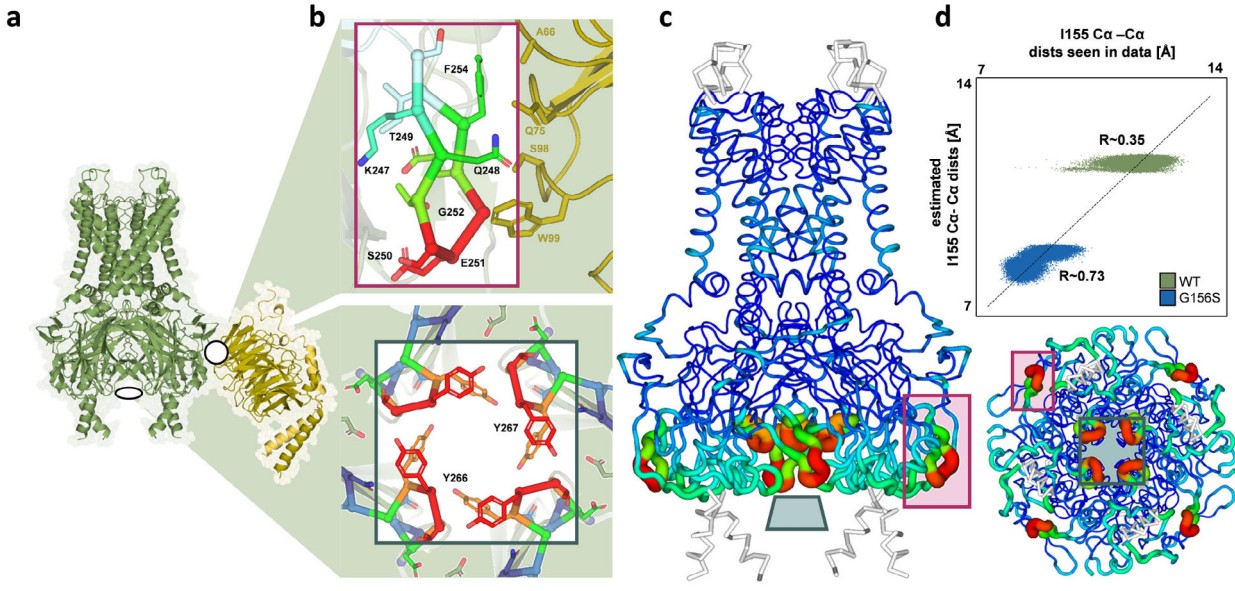

**Fig. 7 Functional mode analysis (FMA) of the aberrant SF fluctuations at I155 of mGIRK2$_{wv}$. a** Gβγ-bound mGIRK2$_{wt}$ according to the 4KFM crystal structure. mGIRK2 is shown in green, one Gβγ subunit is shown in shades of orange. **b** Close-ups of the two regions, which were seen to be most affected in the FMA model. The top figure depicts a side view of the Gβγ-binding site. The lower figure shows a bottom-up view of the inner loops of the CTD. The colors of the residues correspond to the colors allocated in the FMA model in (**c**). **c** Correlations predicted by the FMA model projected on the side (left) and on the bottom-up (bottom right) view of mGIRK2. The backbone of mGIRK2 is shown in rainbow colors. Red indicates a high correlation of movements in the respective region to the Cα distance fluctuation of I155, while blue indicates no correlation. White parts were not considered in FMA model calculation. The close-ups shown in b are indicated with purple and cyan rectangles for the Gβγ binding site and the inner loops of the CTD, respectively. **d** Validation plot of the FMA of MD simulations with the mGIRK2$_{wt}$ and mGIRK2$_{wv}$ channel.

CTD is a feature unique to mGIRK2$_{wv}$. However, it has to be mentioned that the mGIRK2$_{wt}$ SF is much less dynamic than the mutant SF in our simulations. The distance fluctuations of I155 in the wild-type SF might be too small to be linked to any other movements, hence hampering the detection of coupled collective motions. For mGIRK2$_{wt}$, future simulations with Gβγ-bound GIRK2, which go beyond the scope of this study due to the inherent computational effort, might be necessary to make traces of a coupling between the SF and the Gβγ-binding site visible.

**Ba$^{2+}$ block as sensor to test Gβγ-induced changes in SF dynamics.** Ba$^{2+}$ is a high-affinity pore blocker of inwardly rectifying K$^+$ channels, including GIRK[23,27–29], which blocks the pore by interacting with the SF[29,38] mainly at site 4[39]. Motivated by our computational findings, we deemed it possible that Ba$^{2+}$ may serve as a sensor of Gβγ-induced changes in SF dynamics. To test this hypothesis, we measured the parameters of Ba$^{2+}$ block of hGIRK2$_{wt}$, in oocytes that expressed the channel alone, or hGIRK2$_{wt}$ with Gβγ. Ba$^{2+}$ blocked the basal hGIRK2$_{wt}$ currents in a dose-dependent manner with an IC$_{50}$ of 26.4 ± 1.4 μM (Fig. 8). In the presence of Gβγ, the affinity of block was about 2.5 times lower, the IC$_{50}$ was 66.9 ± 8.2 μM. These findings are in line with our hypothesis of a coupling between the SF and the Gβγ-binding site, as suggested by our MD simulations.

## Discussion
Gating of K$^+$ channels occurs in response to channel-specific stimuli, which in the context of K$_{ir}$ channels is facilitated by PIP$_2$,

as well as various additional regulators, such as Na$^+$, Gβγ, ethanol, ATP, and sulfonylurea receptor subunits. Most of these ligands do not interact with the pore domain directly, but induce gating changes by binding to regulatory domains. In case of GIRK channels, Na$^+$ and Gβγ bind to different regions at the CTD, which leads to physical changes at the gates and to a release of steric constrictions impeding K$^+$ flux.

Previous studies[5–10,40–42] have identified ligand-dependent conformational changes important for GIRK2 activity and suggest a complex allosteric network connecting the gates and the modulatory binding sites. Consensus exists about the relevance of large scale movements, such as rotation and rocking of the CTD and bending of transmembrane helix 2 for channel activity, while several aspects about the detailed coupling underlying channel opening have been revealed[6–11].

In an effort to gain a fresh perspective on GIRK2 regulation, we choose an alternative approach to shed light on the allosteric consequences of Gβγ binding by making use of the well-studied *weaver* mouse mutant (mGIRK2$_{wv}$)[14–20,43], which carries a mutation in the SF. Importantly, this mutation does not only drastically alter channel selectivity, but produces an aberrant basal inward current that lacks G-protein activation[14,16,19].

Analysis of μs-long MD simulations suggest a coupling between the SF and the Gβγ-binding site in mGIRK2$_{wv}$. More specifically, functional mode analysis (FMA) unveiled a correlation of aberrant SF dynamics at residue I155 with movements of residues previously reported to contact Gβγ[6] (Fig. 7). Thus, our computational results indicate that mutation-induced changes in the SF are translated to conformational changes in the

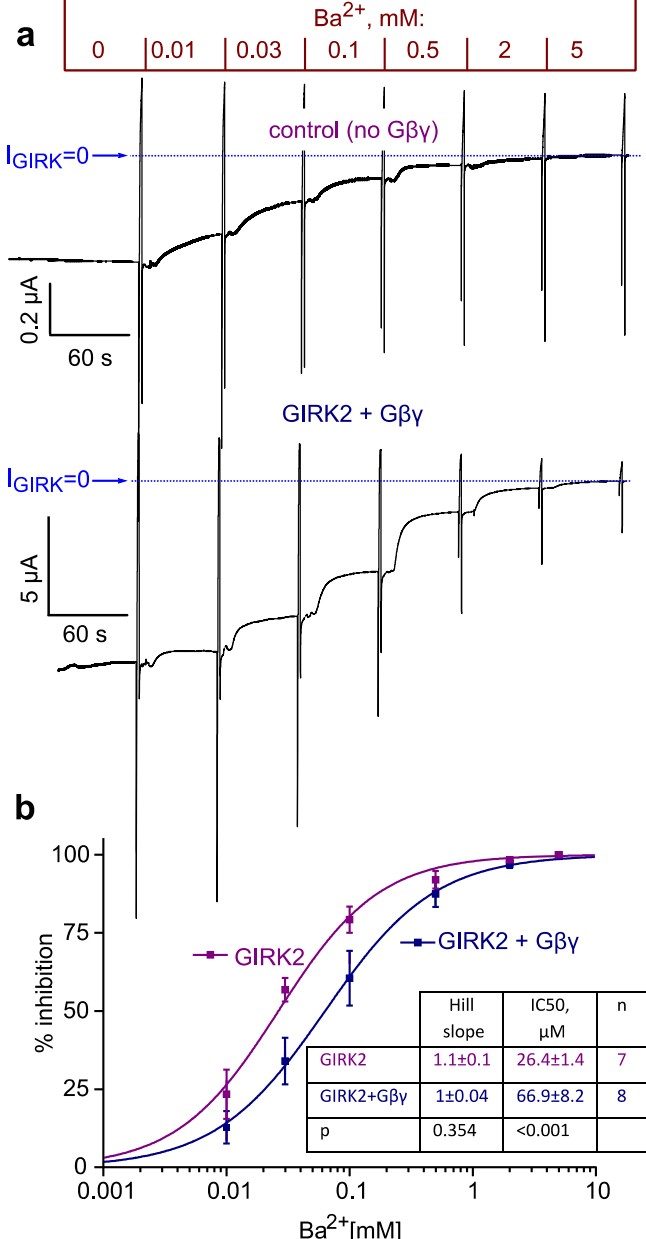

**Fig. 8 Ba$^{2+}$ block is altered following activation of hGIRK2$_{wt}$ by G$\beta\gamma$.**
**a** Representative records of Ba2+ dose–response experiments in oocytes expressing hGIRK2$_{wt}$ (upper trace) or hGIRK2$_{wt}$ + G$\beta\gamma$ (lower trace). The currents were measured at −80 mV. A voltage ramps was applied toward the end of each episode with the indicated Ba$^{2+}$ concentration, and after that the solution was switched to the next, higher Ba$^{2+}$ concentration. **b** Summary of the experiment. Averaged normalized data (% of current inhibition) are shown as mean ± SEM. These averaged dose–response relationships were fitted to the Hill equation (Eq. (9), see the "Methods" section) with the following parameters: hGIRK2, $h$ (Hill slope) = 1.053, IC$_{50}$ = 26 μM; hGIRK2 + G$\beta\gamma$, $h$ = 0.98, IC$_{50}$ = 63 μM. In addition, in each oocyte, the dose–response relationship was fitted separately to the Hill equation. Average parameters ($h$ and IC$_{50}$) are shown in the table (inset). Statistical analysis was done using $t$-test (Hill slope) and Mann–Whitney test (IC$_{50}$).

G$\beta\gamma$-binding site. To further test this correlation, we performed electrophysiological experiments with the human GIRK2$_{G154S}$ channel, which carries a disease-causing mutation corresponding to the *weaver* mutation. We found that, as expected, hGIRK2$_{G154S}$

channels are constitutively active and cannot be further activated by G$\beta\gamma$. Therefore, hGIRK2$_{G154S}$ displays a behavior reminiscent of mGIRK2$_{wv}$ and is likely to experience a similar coupling between SF and the G$\beta\gamma$ regulatory site.

Ba$^{2+}$ is a high-affinity blocker of K$^{+}$ channels including GIRK[23,27–29], which blocks the pore by interacting with the SF[29,38] mainly at site 4[39]. Given the location of the binding site in the SF, we deemed it possible that Ba$^{2+}$ serves as a sensor of G$\beta\gamma$-induced changes in SF dynamics. Our electrophysiological experiments show that G$\beta\gamma$ activation of hGIRK2$_{wt}$ is accompanied by a ~2.5-fold decrease in the affinity of Ba$^{2+}$, compared to hGIRK2$_{wt}$ alone. Importantly, this finding corroborates the presence of an allosteric link between the SF and the G$\beta\gamma$-binding site in wild-type hGIRK2 channels, as well.

The concept of allosteric networks connecting diverse gating regions, including the SF, and modulatory sites is well-established in the field of K$^{+}$ channel regulation. For example, conformational changes in the SF, which affect channel conductivity, have been observed to allosterically influence an inner gate in MthK[44], KcsA[45] and NaK[46] channels. However, much less is known about the interplay between ligands binding to intracellular domains and gating at the SF. One such coupling was observed for the K2P channel TREK-2, which is allosterically activated at the SF by binding of the small molecule 2-Aminoethoxydiphenyl borate to the proximal C-terminus[47].

Recently, Chen et al.[48] suggested coupling between the SF and the G$\beta\gamma$-binding site of mGIRK2 based on electrophysiology experiments that revealed mGIRK2$_{wv}$ stimulation by the channel activator Ivermectin, but not the G$\beta\gamma$-mediated pathway. Furthermore, Li et al.[42] described the GIRK2 SF as a crucial determinant for K$^{+}$ permeation, which can be influenced by binding of G$\beta\gamma$ and Na$^{+}$. In line with these studies, our findings demonstrate a coupling between the GIRK2 SF and the G$\beta\gamma$ binding site corroborated by both experimental and computational data. Against this background, we put forward a new aspect of GIRK2 gating by extending the role of the SF from a mere energy barrier to a G$\beta\gamma$-controlled gating region, thereby adding it to the classical gates controlling K$^{+}$ permeation.

MD simulations with mGIRK2$_{wv}$ show that the mutation G156S leads to aberrant movements and an altered geometry of the SF (Figs. 5b, c and 6), which are likely to affect inherent features of the ion channel, such as selectivity and conductivity. Moreover, the mutation might alter the coupling between the SF and G$\beta\gamma$-binding site, thereby providing a possible explanation for the impaired mGIRK2$_{wv}$ activation by G$\beta\gamma$ observed in experiments[14,17,19]. However, it is unclear whether the mutation changes the affinity of modulators binding to the G$\beta\gamma$-binding site, such as ethanol or the G$\beta\gamma$ subunit. This aspect remains to be elucidated in future studies, since the required modeling is beyond the scope of this framework.

Given the fact that the SF has been observed to couple to channel gating regions in a number of different K$^{+}$ channels[44–46], we analyzed the effect of the SF mutation on the HBC gate by measuring the minimum distances between opposing F192 residues as well as F192 Cα atoms. In addition, previous studies on GIRK channels provided evidence for an allosteric control of the HBC gate by G$\beta\gamma$ binding at the CTD[9,49]. Aiming to clarify the role of the HBC gate in the coupling between the SF and the G$\beta\gamma$ binding site, FMAs were carried out to identify movements in the channel correlated to the motions of the HBC gate. At this point it should be stressed that a conductive state of mGIRK2$_{wt}$ was used as starting point of the simulations and conclusions drawn from these analyses are hence limited to the open HBC gate. Distances at the HBC gate region did not differ in simulations with mGIRK2$_{wt}$ and mGIRK2$_{wv}$ (Supplementary Fig. 3), and we could not identify correlations between HBC gate motions and

other movements within the channel. Thus, our data does not provide evidence that the open HBC gate is affected by the SF mutation or involved in the coupling between the SF and the Gβγ binding site. Further investigations are needed to clarify these aspects for other HBC gate conformations and dynamics.

MD simulations performed in this study do not take into account the role of the CTD conformation in GIRK2 gating, as it was recently suggested by Cryo-EM structures[10]. More precisely, the CTD was reported to switch from an extended to a docked conformation upon $PIP_2$ binding, which makes GIRK2 susceptible to Gβγ binding. Since modeling of the different CTD conformations is beyond the scope of this study, it remains to be elucidated how the coupling between the SF and the Gβγ-binding site is influenced by conformational changes of the CTD, or whether conformational changes of the CTD themselves affect the SF. However, the MD simulations in this framework were carried out with the $PIP_2$-bound mGIRK2 channel, which, according to the mechanism proposed by the Cryo-EM structures, adopts the docked conformation. Under these conditions, we do not expect a conformational change of the CTD. Thus, the coupling between the SF and the Gβγ-binding site may be considered valid for the $PIP_2$-bound channel, independent of conformational changes upon $PIP_2$ removal.

In addition to addressing structural–biological questions, our work strives to investigate the SF mutation from a pharmacological perspective. The human $GIRK2_{G154S}$ channel carries a mutation corresponding to the *weaver* mutation G156S in mGIRK2 and causes the rare and severe neurological Keppen–Lubinsky syndrome (KPLBS)[13]. We here present the functional and structural characterization of the human $GIRK2_{G154S}$ disease mutant, which we show to exhibit properties strongly reminiscent of $mGIRK2_{wv}$, including loss of inward rectification, acquisition of sensitivity to block by the sodium channel blocker QX314, high constitutive activity and reduced $K^+$ selectivity[14,16,19]. In agreement with our electrophysiology measurements, mutant channels are permeable to both $K^+$ and $Na^+$ in our simulations. $Na^+$ ions permeate in a partly hydrated way, while $K^+$ sheds its water shell upon entry into the SF. Considering the substantial energy barrier associated with $Na^+$ dehydration, the exclusion of water is likely to prevent the passing of $Na^+$ flux though $mGIRK2_{wt}$. A similar mechanism has been reported for NaK channels[50], which narrow and dilate the SF in order to enable the permeation of dehydrated $K^+$ and partially hydrated $Na^+$ ions, respectively.

The replacement of G156 with a serine in $mGIRK2_{wv}$ causes a destabilization of the SF and enables the S156 side chain to form a hydrogen bond with the backbone oxygen of I155. Hydrogen bonds surrounding the SF have been observed to form a delicate network in KcsA, which, if altered, influences the energy landscape of the filter and simultaneously the behavior of the channel[51]. Reminiscent of this phenomenon, the formation of the hydrogen bonds between I155 and S156 is accompanied by fundamental changes in the $mGIRK2_{wv}$ SF. Firstly, I155 can adopt an additional conformation, which is reflected in a bimodal distribution of distances between the I155-Cαs of opposing subunits. Secondly, S156 has to rotate in order to participate in the hydrogen bond, which leads to a shift in the phi backbone angle of this residue. The so-called "flipping" of SF residues, which corresponds to a rotation of the SF backbone away from the cavity and a concomitant fluctuation of the torsion angles of the affected SF residues, has previously been associated with MthK inactivation[44]. The altered dynamics additionally lead to the entry of water molecules behind the SF region, which, together with the carbonyl flipping, was identified as crucial factor for SF gating of KcsA[51] and MthK[52] channels.

$Na^+$ ion occupancies in $mGIRK2_{wv}$ do not show any considerable peaks suggesting a rather accelerated flux and therefore decreased energy barriers for $Na^+$ permeation. There are two main occupancy sites of $K^+$ in $mGIRK2_{wv}$: the first one coincides with the site S3, which was also populated in $mGIRK2_{wt}$, while the second binding site is between S1 and S2. Importantly, both S1 and S2 are flanked by the mutated residue S156, whose deviating dynamics are likely to play a crucial role in shifting $K^+$ occupancy from S2 to an in-plane position. Studies on several types of $K^+$ channels directly linked ion occupancy to changes in SF conformation and gating[44,52–55]. For example, high ion occupancies steered by the external $K^+$ concentration have been associated with increased SF stability on MthK[44]. Remarkably, $K^+$ occupation have been reported to be crucial for MthK[44,52] and KirBac[56] channel regulation.

We here present a mechanism underlying GIRK2 activation, which is coined by an allosteric link of the Gβγ-binding site to the SF. This mechanism complements previously proposed hypotheses about GIRK2 regulation by suggesting the SF as an additional gate controlling $K^+$ permeation. We believe that our model may be extendable to ligand-dependent modulation of other $K^+$ channels, since a similar mechanism has been reported for K2P channel activation[47]. Furthermore, our study is of relevance to human disease, as we present the first-time functional characterization of the human $GIRK2_{G154S}$. Together with our computational findings, this study will lead to a better understanding of the associated syndrome, KPLBS, and prepare the ground for therapeutic approaches targeting KPLBS and other GIRK-related diseases.

## Methods

**Ethical approval of Xenopus laevis and oocyte preparation**. Oocyte experiments have been approved by Tel Aviv University Institutional Animal Care and Use Committee (permits # 01-16-104 and 01-20-083). Maintenance and surgery of female *Xenopus* laevis frogs were as described previously[57]. Frogs were housed in dechlorinated water tanks, on a 10 h light/14 h dark cycle at $19 \pm 2°C$. Portions of ovary were removed under anesthesia with tricaine methanesulphonate (0.17%). After suturing the incision, frog was held in a separate container to fully recover from the anesthesia and then shifted to post-operational animals' tank. The frogs did not show any signs of post-operational distress and were allowed to recover for at least three months. After three to four surgeries, anaesthetized frogs were killed by decapitation and double pithing. Oocytes were defolliculated by collagenase, injected with RNA and incubated for 2–3 days at 20–22 °C in ND96 solution (low $K^+$) (in mM: 96 NaCl, 2 KCl, 1 $MgCl_2$, 1 $CaCl_2$, 5 HEPES, pH 7.5), supplemented with 2.5 mM sodium pyruvate and 50 µg ml$^{-1}$ gentamycin[30].

**DNA constructs and RNA**. DNA constructs—hGIRK2, $hGIRK2_{G154S}$, bovine Gβ$_1$, and bovine Gγ$_2$ were cloned into high expression oocyte vectors pGEM-HE or pGEM-HJ as described previously[58,59]. cDNA construct of hGIRK2 was kindly provided by the Tel Aviv University Blavatnik Center for Drug Discovery, in pCDNA3 vector and subcloned into pGEM-HJ vector. The predicted protein sequence corresponds to GenBank accession #NM_0O2240. PCR-site directed mutagenesis was performed on hGIRK2-pGEM-HJ to generate $hGIRK2_{G154S}$-pGEM-HJ. The mutation was confirmed by sequencing the full-length cDNA of this construct. In vitro RNA synthesis and microinjection was performed as described previously[30]. The amounts of RNA injected per oocyte were: hGIRK2 (1 ng) or $hGIRK2_{G154S}$ (0.5 ng), bovine Gβ$_1$ (5 ng), and Gγ$_2$ (1 ng).

**Electrophysiology**. Whole-cell hGIRK currents were measured using the standard two-electrode voltage clamp method at 20–22 °C, in high-K (HK) solution containing 8, 24, or 72 mM extracellular $K^+$ concentration, $[K^+]_o$[26]. These solutions were obtained by mixing ND96 with a 96 mM $K^+$ solution containing, in mM: 96 KCl, 2 NaCl, 1 $CaCl_2$, 1 $MgCl_2$, 5 HEPES, pH adjusted to 7.5 with KOH. Chemicals used in this study are listed in Supplementary Table 2. QX314 chloride was dissolved in water to a final concentration of 1 M. To study QX314 block of hGIRK2, the drug was diluted in the extracellular solutions to a final concentration of either 400 µM or 1 mM. Oocytes expressing the $hGIRK2_{G154S}$ channels did not survive the 2-day incubation period in standard ND96 solution, presumably because of the continuous depolarization caused by $Na^+$ influx via the expressed channels. Therefore, incubation of $hGIRK2_{G154S}$-expressing oocytes was done in the presence of 400 µM QX314, until recording. QX314 was washed out with ND96 after placing

the oocyte in the recording chamber, and the experimental protocol was started after a complete washout that took about 4–5 min. To rule out any effect from QX314 incubation, all the other groups under study, including the wild-type (WT) hGIRK2, were also subjected to QX314 incubation till the day of experiment. Oocytes expressing hGIRK2 channels and naïve oocytes were placed in a dish with ND96 solution for 4–5 min to wash off the QX314 before transferring them to recording chamber.

Current–voltage ($I$–$V$) relations were obtained using 2 s voltage ramps from −120 to +50 mV. For hGIRK2$_{wt}$, net hGIRK $I$–$V$ relationships were obtained by subtracting the current remaining after blocking all hGIRK activity with 1 mM Ba$^{2+}$ [26] (Fig. 3c and Supplementary Fig. 2b–d). The net $I$–$V$ curves for hGIRK2$_{G154S}$ were obtained by subtracting the averaged $I$–$V$ curves obtained from naïve oocytes of the same batch from the QX-sensitive currents (Fig. 3e and Supplementary Fig. 2a, e, f).

**Data analysis**. $I$–$V$ curves were analyzed using Clampfit 10.7 software (Molecular Devices). The reversal potential ($V_{rev}$) was determined from the intercept of the net $I$–$V$ curve with the voltage axis. The extent of inward rectification ($F_{ir}$) was determined by dividing the current at 50 mV positive to $V_{rev}$ by the current at 50 mV negative to $V_{rev}$ (see Fig. 4e)[60]:

$$F_{ir} = \frac{I_{V_{rev}+50}}{I_{V_{rev}-50}} \quad (1)$$

Estimated reversal potential ($V_{rev}$) was calculated using the Nernst equation:

$$V_{rev} = \frac{RT}{zF}\ln\left(\frac{[X]_{out}}{[X]_{in}}\right) \quad (2)$$

Permeability ratio, $r$, of sodium to potassium ($p$Na$^+$/$p$K$^+$) was determined from Goldman–Hodgkin–Katz equation:

$$E_{rev} = \frac{RT}{zF}\log\left(\frac{p_{K^+}\left[K^+\right]_o + p_{Na^+}\left[Na^+\right]_o}{p_{K^+}\left[K^+\right]_i + p_{Na^+}\left[Na^+\right]_i}\right) \quad (3)$$

For the conditions of our experiments, performed at 22 °C, we used $RT/zF = 58.17$ mV. Since the intracellular concentrations of K$^+$ and Na$^+$ are not exactly known, $r$ was determined from measurement of $E_{rev}$ ($V_{rev}$) in two extra-cellular solutions containing different K$^+$ and Na$^+$ concentrations, [K$_o$](1) and [Na$_o$](1) and [K$_o$](2) and [Na$_o$](2), for the pair of K$^+$ external concentrations of 8 and 72 mM, as follows:

$$V_{rev} = 58.17\log\left[\frac{[K^+]_o + r[Na^+]_o}{[K^+]_i + r[Na^+]_i}\right] \quad (4)$$

For two combinations of Na$^+$ and K$^+$ in external solution, 1 and 2, and with the corresponding $V_{rev}$ measured in these solutions:

$$\left[\frac{[K^+]_o(1) + r[Na^+]_o(1)}{[K^+]_o(2) + r[Na^+]_o(2)}\right] = \left[\frac{10^{\frac{V_{rev}(1)}{58.17}}}{10^{\frac{V_{rev}(2)}{58.17}}}\right] \quad (5)$$

Denoting $\left[\frac{10^{\frac{V_{rev}(1)}{58.17}}}{10^{\frac{V_{rev}(2)}{58.17}}}\right]$ as A

$$[K^+]_o(1) + r[Na^+]_o = A[K^+]_o + rA[Na^+]_o(2); \quad (6)$$

$$r\left[[Na^+]_o(1) - A[Na^+]_o(2)\right] = A[K^+]_o(1) \quad (7)$$

$$r = \frac{A[K^+]_o(2) - [K^+]_o(1)}{[Na^+]_o(1) - [Na^+]_o(2)} \quad (8)$$

In each oocyte, the dose–response curves for Ba$^{2+}$ inhibition of hGIRK currents were fitted to Hill equation in the form

$$\%\text{inhibition} = \frac{100\%x^h}{EC_{50}^{\ h} + x^h} \quad (9)$$

where $x$ is the extracellular concentration of Ba$^{2+}$, and $h$ is Hill coefficient (slope).

**Statistical analysis**. Statistical analysis was performed using SigmaPlot 13 (Systat Software, Inc.) and GraphPad Prism version 8 for Windows (GraphPad Software, La Jolla, CA, USA). If the data passed the Shapiro–Wilk normality test and the equal variance test, two-group comparisons were performed using $t$-test, while multiple group comparison was performed using one-way ANOVA. If the data were not distributed normally, the Mann–Whitney rank sum test, or Kruskal–Wallis test were performed, respectively.

**Statistics and reproducibility**. All electrophysiological measurements were taken from single oocytes. The electrophysiology data are reported as mean ± SEM if the data in all test groups passed the Shapiro–Wilk normality test and the equal variance test. In most figures the results from individual cells are also shown within bars. If the data did not pass this test they were presented as box plots showing individual data points and 25th and 75th percentiles (Fig. 4d). Statistical analysis was performed using SigmaPlot 11 or 13 (Systat Software, Inc.) and GraphPad

Prism version 9 for Windows (GraphPad Software, La Jolla, CA, USA). "Before–after" tests of an effect of a substance measured in single cells (Fig. 3f) were analyzed using paired $t$-test followed by Wilcoxon test. Two-group comparisons were performed using $t$-test if the data passed the Shapiro–Wilk normality test and the equal variance test, otherwise we used the Mann–Whitney Rank Sum Test. Multiple group comparisons were done using one-way ANOVA (ANOVA on ranks was performed whenever the data did not distribute normally). Tukey's or Dunnett's tests were performed for normally distributed data and Dunn's or Kruskal–Wallis test otherwise. Statistical differences are reported in figure legends.

**Molecular dynamics simulation**. Gromacs (version 5.1.2)[61] was used to perform molecular dynamics simulations with WT and G154S K$_{ir}$ 3.2 channels (PDB code: 3SYA; resolution 2.98 Å, Organism: *Mus musculus*)[5]. As starting conformation, we extracted a frame of one of our previous simulations, in which mGIRK2 is in a conductive state[7]. The channels were embedded in an equilibrated 1-palmitoyl-2-oleoyl-sn-glycero-3-phosphocholine (POPC) membrane, using g_membed. Berger lipid parameters[62] were used for lipids and the amber99sb forcefield[63] for the pro-tein. PIP$_2$ parameters are described in our previous work[64]. The system was solvated with 65,750 SPC/E water molecules[65]. A total salt concentration of ~180 mM was added to the simulation system (90 mM KCl, 90 mM NaCl). The corrected monovalent Lennard–Jones parameters[66] for ions were applied and Lennard–Jones and electrostatic interactions were cut-off at 1.0 nm. Long-range electrostatic inter-actions were calculated every step with the Particle-Mesh Ewald algorithm[67] and bonds constrained with the LINCS algorithm[68], allowing for an integration time step of 2 fs. Temperature was coupled to 310 K using the v-rescale thermostat[69] with a coupling constant of 0.1 ps. The pressure was kept constant semi-isotropically at 1 bar with the Parrinello–Rahman[70] barostat ($\tau = 2$ ps). In the production runs, G-loop gate movement was controlled by restraining the residues G318 and M319 with 1000 kJ mol$^{-1}$ nm$^{-2}$ to their initial positions. During energy minimization protein atoms were restrained with a force constant of 1000 kJ mol$^{-1}$ nm$^{-2}$ to their starting positions. The minimization was done by using the steepest descent algo-rithm, followed by 10 ns of NVT and 10 ns of NPT equilibration runs.

Six runs of mGIRK2$_{wt}$ and mGIRK2$_{wv}$ were calculated resulting in a total simulation time of 6 µs for the wild-type and 9 µs for the mutant system (Supplementary Table 1). The runs were carried out under different electric fields of 40, 50, and 60 mV nm$^{-1}$ along the z-axis. Taking into account the thickness of the membrane of ~3.5 nm, the electric fields yield transmembrane potentials of 140, 175, and 210 mV, respectively.

Analysis of distances and angles was conducted using Gromacs. In order to obtain average distances, the distances between opposing residues were determined and the mean of the two values calculated. The average distance between opposing I155 Cαs, the average distance between F192 Cαs as well as the minimum distances between F192 were considered as functional quantities for the respective functional mode analysis (FMA) models. FMA was carried out with the FMA tool developed by Hub and de Groot[37]. As input for the FMA, we combined all frames of the MD simulation with mGIRK2$_{wt}$ and mGIRK2$_{wv}$ to obtain a continuous 6 and 9 µs trajectory, respectively. We performed a principle component analysis of the protein backbone and used the first 50 principle component vectors as a basis set. In order to reduce background noise, we excluded the flexible extracellular loop region (residue 120–133) and C-terminus (residue 369–382) from our model. Different allocations to model-building and cross-validation set were probed to find the division of the data, which yields the best models. For mGIRK2$_{wv}$, the first 2.5 µs were the basis for model building, while the last 6.5 µs were used for cross validation. For mGIRK2$_{wt}$ the model was trained with the first 5 µs and validated with the last 1 µs. Correlation was quantified with both Pearson's $R$ as well as the mutual information. Visualization of the model was based on the ensemble-weighted maximally correlated motion.

**Reporting summary**. Further information on research design is available in the Nature Research Reporting Summary linked to this article.

## Data availability
MD simulation data and source data underlying figures that support the findings of this study have been deposited in a Zenodo repository (https://doi.org/10.5281/ZENODO.6375552)[71]. Further data supporting the findings of this paper are available from the corresponding authors upon reasonable request.

## Code availability
Results of electrophysiology experiments were analyzed using the Clampfit 10.7 software (Molecular Devices). Statistical analysis was performed using SigmaPlot 13 (Systat Software, Inc.) and GraphPad Prism version 8 for Windows (GraphPad Software, La Jolla California USA). Simulation trajectories were collected using the simulation program GROMACS. Visualization and analysis were performed using VMD, Pymol, Python, Java and the FMA tool. All of these software packages are publicly available.

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

## Acknowledgements
The computational results presented have been achieved in part using the Vienna Scientific Cluster (VSC). This work was supported by the doctoral program "Molecular drug targets" W1232 (T.F. and A.S.-W.) and the Post-Doc program "Zukunftskolleg" ZK-81B of the Austrian Science Fund (FWF) (T.F. and E.-M. Z.-P.), the DOC fellowship 26156 of the Austrian Academy of Sciences (ÖAW) (T.F.), the Israel-India Binational grant (Israeli # ISF_2255_2015 and Indian # UGC-6-1/2016 (1C) (N.D. and A.K.B); Israel Science Foundation grant # ISF_1282_2018 (N.D.).

## Author contributions
A.S.-W., T.F., and H.B. conceived the computational design of the project. T.F. and H.B. carried out molecular modeling, which was analyzed by A.S.-W., T.F., H.B., and E.-M.Z.-P. Experiments were designed by N.D. and H.P.R. and carried out by H.P.R., B.S., and J.C.C. A.S.-W., T.F., N.D., and H.P.R. wrote the paper, with contributions from H.B., A.K.B., and E.-M.Z.-P.

## Competing interests
The authors declare no competing interests.
