## [Peer Review File · Communications Biology]

Reviewers' comments:

Reviewer #1 (Remarks to the Author):

In this study Friesacher and Reddy, from the Stary-Weinzinger and Dascal labs in Vienna and Tel Aviv respectively, explore the molecular mechanism of action of a Gly selectivity filter (TIGYGYR) mutant of the GIRK2 channel that causes disease. This mutation has long been known to underlie the disease symptoms in the weaver mouse, while more recently it has been associated in humans with the rare but severe neurological disorder called the Keppen-Lubinsky syndrome. In the mouse, this glycine to serine mutant has been characterized by a loss of K⁺ selectivity to Na⁺ permeability, elevated basal activity but impaired activation by the Gβγ dimer and insensitivity to extracellular Ba²⁺ block.

The authors perform long MD simulations of the weaver mutant (9 μs) and the wild-type (6 μs) channels establishing in their simulation results existing experimental results: 1) they show a dramatic reduction of selectivity to K⁺ over Na⁺ ions by the GIRK2G156S mutation (pK/pNa from 40 to ~3) concomitant with a drop in the K⁺ permeation rate (from 6.7 to 1.4 ions/μs) and an increase in the Na⁺ permeation rate (from 0.16 to 2.25 ions/μs); 2) In the wild-type in 3/6 runs, Na⁺ ions blocked K⁺ ion flow getting stuck in sites S2 or in sites S3 and S4, while in the weaver mutant water was not excluded from the SF as in the wild-type and thus Na⁺ ions could permeate. These results are reassuringly consistent with previous experimental results, validating the computational approach.

Similarly, the experimental approach shown in Figs. 4 and 5 recapitulated characteristics of human wild-type and GIRK2G154S channels such as 1) the dependence of the wild-type activity (V_m-E_K and conductance) on [K⁺]_o but not of the mutant; 2) block by the cation channel blocker QX314 of the mutant but not the wild-type; 3) block of the wild-type by Ba²⁺ but not of the mutant; 4) stimulation of currents by Gβγ of the wild-type but not the mutant; and 5) greater rectification of the wild-type compared to the mutant.

Novel information is presented in Figs. 5-8. Three computational figures provide mechanistic information on a) how the mGIRK2G156S hydrogen bonds with I155 and permeating water molecules, while the wild-type G156 fails to do so; b) not only differences in the position of the Gly and Ser residues at position 156 could be seen between wild-type and mutant channel but also in the flanking residues (I155 and Y157) as well as in T154; c) using functional mode analysis the authors show correlated movements between I155 and βD-βE loop, one of the two critical loops where Gβγ binds. This last novel result is further reinforced by a novel experimental finding showing in Fig. 8 that Ba²⁺ block, which is known to get stuck at the S4 site, works less well (IC₅₀ is right-shifted) when Gβγ is co-expressed, suggesting that the presence of Gβγ (75 Å away) is felt at the selectivity filter. Both theory and experiment are performed well with the appropriate controls. This is a study well-done combining theory and experiment in a meaningful way about a disease mutant. The suggestions below are intended to improve the paper further.

Major point

Given that the point made by the last two figures is the most novel one both computationally and experimentally, it would strengthen the study to include in the computational analysis the helix bundle crossing (HBC) gate. HBC has been implicated to be the predominant gate opened by Gβγ by both theoretical studies (e.g. by ref. #9 - Li et al., 2019 - that suggests that it causes a rocking movement that opens predominantly this gate) as well as experimental ones (e.g. Jin et al., Molecular Cell 2002, 10:469-481). Furthermore, multiple studies have provided evidence that the SF is coupled with the gates of K⁺ channels. Thus, the question becomes whether the Gβγ coupling to the SF proceeds via the HBC. Either way the results turn out, it will inform mechanistically the conclusion that the βD-βE loop and SF movements are correlated. Similarly, the hGIRK2(S179P) mutant (corresponding to the GIRK4-S176P mutant in the Jin et al., 2002 study) that mimics Gβγ signaling ought to show reduced Ba²⁺ block but not reduced tertiapin Q block (a bee-venom toxin, plugging the pore extracellularly).

Minor points

- 1) Lines 30-32 in the abstract, I would reword the sentence as follows: "This coupling is corroborated by electrophysiological experiments, revealing a dependence of the GIRK2WT Ba²⁺ block in the presence of Gβγ".
- 2) I do not agree with the authors in the way they have presented prior mechanistic work in the field making it appear highly controversial (lines 58-67). Even if there are differences in the conclusions drawn by different studies, given the state of the art at the time the studies were performed, unless the authors deal directly to address such differences it would be best to make a general statement such as "a consensus on how GIRK channels are gated has not yet been reached", and leave it at that.
- 3) I would also refrain from overstatements, such as the one made on lines 83-84: "We provide unprecedented insight into the functional, dynamic and allosteric consequences of the SF mutation".
- 4) Figure 2 ought to be better explained in the text, especially the right panel showing the water flux through the two channels that are being compared. Remember that the paper will be read by both theoreticians and experimentalists, so you need to be cognizant of this and guide the non-expert reader by the hand.
- 5) Lines 159-161, please delete reference to Fig. 3e, as it does not show Ba²⁺ block. Sentence ought to read as follows: "I-V curves of GIRK2G154S with or without Gβγ showed little rectification and were not blocked by Ba²⁺ (Supplementary Fig. 2f)".
- 6) Lines 294-295, I believe the authors meant to cite Fig. 7 not Fig. 8. The sentence ought to read: "Functional Mode Analysis (FMA) unveiled a correlation of aberrant SF dynamics at residue I155 with movements of residues previously reported to contact Gβγ6 (Fig. 7)".
- 7) Lines 332-333, I would replace the word "alleviate" with "prevent" so that the sentence reads: "Considering the substantial energy barrier associated with Na⁺ dehydration, the insufficient exclusion of H₂O is likely to prevent the passing of Na⁺ flux through GIRK2WT".
- 8) Lines 477-480: "The runs were carried out under different electric fields of 40 mV nm⁻¹, 50 mV nm⁻¹ and 60 mV nm⁻¹ along the z-axis. Taking into account a z-axis box length of ~14.5 nm, the electric fields yield transmembrane potentials of ~580 mV, ~725 mV and 870 mV, respectively." I believe this is not correct. The thickness of the membrane should be used (about 3.5nm) rather than the length of the simulation box. The transmembrane potentials should thus be about 140mV, 175mV, and 210mV for the simulations.
- 9) Dr. Dascal should complete the "Acknowledgements" and "Author Contributions" sections.
- 10) I would indicate either mGIRK2 for mouse or hGIRK2 for human to always be clear which channel is being used.

Reviewer #2 (Remarks to the Author):

The GIRK2 channel mouse weaver mutation has been studied for more than 30 years. Identified in mice, numerous studies have established that the weaver mutation, a Glycine to Serine (G156S) substitution near the selectivity filter, results in profound changes in GIRK channel behavior including: loss of K⁺ selectivity, loss of inward rectification, Gβγ protein insensitivity, as well as several severe neurological deficits resulting primarily from death of Cerebellar granule cells and depletion of dopaminergic neurons in the substantia nigra. In this manuscript Freisacher et al. study the weaver mutation equivalent in human GIRK channels (G154S) which results in the severe, rare, neurological disorder called Keppen-Lubinsky syndrome (KPLBS). The authors conduct molecular dynamics (MD) simulations of GIRK channels containing the weaver mutation and present results here that describe the mechanism by which loss of K⁺ selectivity may occur as well as downstream differences of the C-terminal domain (CTD) in the region where Gβγ binds which are coupled to the aberrant selectivity filter dynamics. Additionally, the human equivalent of the weaver mutation (hGIRK2 G154S) is expressed and biophysically characterized and it exhibits similar characteristics to those of the weaver mutation in mouse.

The manuscript is generally well written and the data are presented clearly. The validity and rigor of the statistical analysis is strong and the methods are well-detailed, consistent with previous work in the field, and should be easily reproducible by other researchers in the field.

While it has long been established that the weaver mutation results in a loss of $G\beta\gamma$ sensitivity, the description of the allosteric effect of the weaver mutation on the dynamics of the in the $G\beta\gamma$ binding site represents a novel, and important, finding. Additional evidence supporting the selectivity filter as a gate is also of interest to the field.

There is one major point and two minor point that should be addressed to improve the manuscript (listed below). Additionally, the manuscript needs to be more carefully edited as there are multiple typos and errors (i.e. Line 112 the wv subscript is missing, Lines 166 & 168- 254 should be 154, etc.).

Major point

1. The major novel findings in this manuscript are the allosteric effects in the CTD caused by the weaver mutation. The authors state that "simulations with $G\beta\gamma$ bound GIRK2, which go beyond the scope of this study due to the inherent computational effort, might be necessary to make traces of a coupling between the SF and $G\beta\gamma$ binding site visible" which is reasonable. However, absent such simulations there is not sufficient experimental evidence to support their MD findings in the manuscript as presented. The authors use the decrease in barium affinity of the wild-type $G\beta\gamma$ bound channel as evidence to support the allosteric coupling of the SF and the $G\beta\gamma$ binding site. Although this is one explanation for these data, it is not the only possible explanation and is not sufficient to support the MD findings.

An additional experiment that would support the MD simulation finding would strengthen this result. Because $G\beta\gamma$ binding affects PIP2 affinity, a decrease in $G\beta\gamma$ binding would be expected to cause a concomitant decrease in PIP2 binding, so the authors could measure PIP2 affinity to support their hypothesis. Alternatively, alcohol binds GIRK2 channels in the same region of the CTD dynamic changes were described in this study. Therefore, one would expect a decrease in alcohol affinity and, in turn, G-protein independent activation by alcohol in the mutant channel. Either of these experiments would provide additional experimental support that would significantly strengthen the findings of their MD simulations.

Minor points

1. The authors make much of the fact that this paper presents the first biophysical characterization of the human GIRK2 G154S mutation and present this as a major advancement in the field. The human and mouse GIRK2 protein sequences are almost identical (depending on the isoform) and there are no differences between these species in the biophysical properties characterized in this study. Furthermore, the MD simulations use structures solved using the mouse GIRK channel, so the import of the findings in this study are predicated on the mechanism being conserved between species. The manuscript would be improved if the human biophysical characterization was described more as confirmatory (which it is), rather than a significant advancement of the field.

2. Although the new GIRK2 cryo-EM structures are mentioned in the introduction, there is no mention of how these new structures, and the novel extended conformation exhibited in the absence of PIP2, might affect the interpretation of how the selectivity filter might couple to the $G\beta\gamma$ binding region. The manuscript would be improved by at least a mention of such an interpretation.

Dear reviewers,

We would like to express our gratitude for your valuable time spent on giving constructive feedback and carefully reviewing our manuscript. The detailed point-by-point responses to the raised points of critique are listed in the table below.

We marked all introduced changes in the manuscript including the suggested improvements. Furthermore, we corrected typos as well as formatting mistakes and made some minor stylistic alterations in the text. Some of the figures were slightly altered in order to adhere to the Journal's figure formatting guidelines. Please find all changed and new figures at the end of the letter.

Reviewer 1		
Point raised	Response	Changes in manuscript
Major point		
Given that the point made by the last two figures is the most novel one both computationally and experimentally, it would strengthen the study to include in the computational analysis the helix bundle crossing (HBC) gate. HBC has been implicated to be the predominant gate opened by G β by both theoretical studies (e.g. by ref. #9 – Li et al., 2019 - that suggests that it causes a rocking movement that opens predominantly this gate) as well as experimental ones (e.g. Jin et al., Molecular Cell 2002, 10:469-481). Furthermore, multiple studies have provided evidence that the SF is coupled with the gates of K ⁺ channels. Thus, the question	We thank reviewer 1 for this insightful comment and agree on the necessity to clarify the role of the HBC gate in the coupling of the SF to the G β binding site. We measured the minimum distances between opposing F192 and between C α atoms of opposing F192 residues over 9 μ s and 6 μ s of GIRK2 _{wv} and GIRK2 _{wt} simulation data, respectively, in order to compare the behavior of the HBC gate in the wt and the mutant channel. The distance distributions did not show any differences in the behavior of the mutant and the wild type channel (Supplementary Fig. 3). Therefore, we do not see any effects of the SF mutation on the HBC gate. We further conducted a set of Functional Mode Analyses (FMA) for the mutant and wild-type channel to identify motions coupled to movements in the HBC gate. The measured distances between opposing F192 residues were used as functional quantities for the respective models. None of the obtained models yielded a correlation measure higher than 0.23. Therefore, we do not see evidence for an involvement of the HBC gate in the coupling between	Lines 270-282: “The HBC gate is not affected by the SF mutation. In order to analyze whether the HBC gate is influenced by the SF mutation, we measured the distances between F192 of opposing subunits. For both mGIRK2 _{wt} and mGIRK2 _{wv} , the distribution of distances between opposing F192 C α atoms culminates at about 18.5 Å, while the peak of the minimum distance between opposing F192 residues is at around 11 Å (Supplementary Fig. 3). In order to identify the motions in the channel maximally correlated with the movements of the HBC gate, Functional Mode Analyses (FMAs) were conducted. FMA models were constructed using the minimum distances between opposing F192 residues and the distances between F192 C α atoms as functional quantities. None of the models obtained for mGIRK2 _{wv} or mGIRK2 _{wt} exceeded correlation measures of 0.23 in cross-validation. Thus, the MD simulations show that the HBC gate is open to a similar degree in mutant and wild-type mGIRK2 and its movements are not correlated to any other motions in the ion channel. However, it should be emphasized that an open state of mGIRK2 was used as starting point of the simulations ⁷ . A

becomes whether the Gβγ coupling to the SF proceeds via the HBC. Either way the results turn out, it will inform mechanistically the conclusion that the βD-βE loop and SF movements are correlated. Similarly, the hGIRK2(S179P) mutant (corresponding to the GIRK4-S176P mutant in the Jin et al., 2002 study) that mimics Gβγ signaling ought to show reduced Ba²⁺ block but not reduced tertiapin Q block (a bee-venom toxin, plugging the pore extracellularly).	the SF and the Gβγ site in on our simulations. However, it should be kept in mind that a conductive state of the channel was used as starting point for the simulations, and the predictions are limited to the open HBC gate. Further simulations on mGIRK2 with a closed HBC gate would be needed to clarify whether the SF mutation has an effect on the closed HBC gate, which are out of the scope of this study. We included our findings in the result and the discussion part. We added a supplementary figure (Supplementary Fig. 3), which provides the distributions of the distances between opposing F192. Furthermore, we made the necessary changes to the method section explaining the HBC gate analysis. Regarding the hGIRK2(S179P) mutation, we would refrain from assuming that changes in SF caused by Gβγ and by an activating mutation necessarily will be similar. The S176P mutation reportedly affects the HBC gate rather than SF (Jin et al. 2002). However, we agree with the reviewer that an investigation of this mutation is interesting in the light of the results presented in this study. Thorough MD simulations may help to resolve this issue, but are out of scope of the present report.	coupling of the SF movement as well as an impact of the mutation on a more closed HBC gate can therefore not be excluded and remains to be elucidated in future studies.” Lines 426-438: “Given the fact that the SF has been observed to couple to channel gating regions in a number of different K⁺ channels⁴⁴⁻⁴⁶, we analyzed the effect of the SF mutation on the HBC gate by measuring the minimum distances between opposing F192 residues as well as F192 Cα atoms. In addition, previous studies on GIRK channels provided evidence for an allosteric control of the HBC gate by Gβγ binding at the CTD^{9,48}. Aiming to clarify the role of the HBC gate in the coupling between the SF and the Gβγ binding site, FMAs were carried out to identify movements in the channel correlated to the motions of the HBC gate. At this point it should be stressed that a conductive state of mGIRK2_{wt} was used as starting point of the simulations and conclusions drawn from these analyses are hence limited to the open HBC gate. Distances at the HBC gate region did not differ in simulations with mGIRK2_{wt} and mGIRK2_{mv} (Supplementary Fig. 3), and we could not identify correlations between HBC gate motions and other movements within the channel. Thus, our data does not provide evidence that the open HBC gate is affected by the SF mutation or involved in the coupling between the SF and the Gβγ binding site. Further investigations are needed to clarify these aspects for other HBC gate conformations and dynamics.” Lines 605-609: “Analysis of distances and angles was conducted using Gromacs. In order to obtain average distances, the distances between opposing residues were determined and the mean of the two values calculated. The average distance between opposing I155 Cαs, the average distance between F192 Cαs as well as the minimum distances between F192 were considered as functional quantities for the respective Functional Mode Analysis (FMA) models.”
---	--	--

		Lines 614-615: "Different allocations to model-building and cross-validation set were probed to find the division of the data, which yields the best models." Lines 617-618: "Correlation was quantified with both Pearson's R as well as the mutual information." Supplementary Fig. 3: listed at the end of this letter
Minor points		
1) Lines 30-32 in the abstract, I would reword the sentence as follows: "This coupling is corroborated by electrophysiological experiments, revealing a dependence of the GIRK2WT Ba ²⁺ block in the presence of Gβγ".	The sentence was altered, we would propose to word it in a clearer way, so that it is also apparent, how Gβγ influences GIRK2 _{wt} block by Ba ²⁺ .	Lines 30-32: "This coupling is corroborated by electrophysiological experiments, revealing that GIRK2 _{wt} activation by Gβγ reduces the affinity of Ba ²⁺ block."
2) I do not agree with the authors in the way they have presented prior mechanistic work in the field making it appear highly controversial (lines 58-67). Even if there are differences in the conclusions drawn by different studies, given the state of the art at the time the studies were performed, unless the authors deal directly to address such differences it would be best to make a general statement such as "a consensus on how GIRK channels are gated has not yet been reached", and leave it at that.	As suggested by reviewer 1, we introduced a more general statement concerning the prior mechanistic work in this field at the beginning of the paragraph. We agree with reviewer 1 that this section makes the different aspects of GIRK2 gating seem unnecessarily controversial, but we think that an overview of the prior mechanistic work might be helpful for the readers to put the results of this study into context. Therefore, we edited the following paragraph with the aim to give an overview of rather than pointing out the differences between the different aspects of GIRK channel control.	Lines 51-63: "So far, diverse aspects of GIRK channel control have been unraveled ⁵⁻¹² , but consensus on a gating model has not been reached yet. For example, X-ray crystallography experiments ^{5,6} suggest that HBC and G-loop gate opening is facilitated by PIP ₂ and Gβγ in a co-dependent manner through a rotation of the C-terminal domain (CTD) and splaying of the inner helices. Interestingly, a crystal structure of GIRK2 bound to all of its activators does not capture a fully-open channel, motivating the authors to propose a dynamic activated state, which fluctuates between non-conductive and conductive conformations ⁶ . Several MD simulation studies observe an influence of PIP ₂ binding on the HBC gate ^{7,8} . Li et al. ⁹ hold Gβγ and Na ⁺ accountable for the opening of the G-loop gate and HBC gate, respectively, while reporting a stabilizing effect of PIP ₂ binding on the open conformation of the gates. Another putative aspect of GIRK channel gating was revealed by Cryo-EM structures ^{10,11} , which indicate that the CTD of the channel can adopt two different conformations: a docked and an

		extended conformation that differ in the compactness as well as distance of the domain to the transmembrane part of the protein.” Line 383-385: “Consensus exists about the relevance of large scale movements, such as rotation and rocking of the CTD and bending of transmembrane helix 2 for channel activity, while several aspects about the detailed coupling underlying channel opening have been revealed ⁶⁻¹¹. ”
3) I would also refrain from overstatements, such as the one made on lines 83-84: "We provide unprecedented insight into the functional, dynamic and allosteric consequences of the SF mutation".	The text was checked for such statements, which were replaced or deleted.	Lines 82-83: “We provide unprecedented insight into the functional, dynamic and allosteric consequences of the SF mutation.” Lines 381-383: “Previous studies ^{5-10,40-42} have identified numerous ligand-dependent conformational changes important for GIRK2 activity and suggest a complex allosteric network connecting the gates and the modulatory binding sites.” Lines 464-466: “Hydrogen bonds surrounding the SF have been observed to form an extremely delicate network in KcsA, which, if altered, influences the energy landscape of the filter and simultaneously the behavior of the channel ⁵⁰.” Line 485-486: “We here present a novel mechanism underlying GIRK2 activation, which is coined by an allosteric link of the Gβγ binding site to the SF. mechanism underlying GIRK2 activation, which is coined by an allosteric link of the Gβγ binding site to the SF.”
4) Figure 2 ought to be better explained in the text, especially the right panel showing the water flux through the two channels that are being compared. Remember that the paper will be read by both theoreticians and experimentalists,	We thank reviewer 1 for this insightful remark. We inserted a more elaborated explanation of the figure in the text and added a more detailed caption for Fig. 2.	Lines 111-120: “Fig. 2 Ion and water flux through the mGIRK2_{wv} (top) and mGIRK2_{wt} (bottom) SF over simulation time. a Representative snapshots of ion and water flux through the SF. mGIRK2_{wv} is colored blue, while mGIRK2_{wt} is colored green. The SF is depicted in cartoon representation with the backbone oxygens of the SF residues indicated as red sticks. K⁺ ions are represented as pink spheres, while Na⁺ ions are represented as

so you need to be cognizant of this and guide the non-expert reader by the hand.		smaller, light blue spheres. Water molecules are shown in a ball-and-stick representation. b K⁺ (purple) and Na⁺ ions (blue) flux observed in runs with mGIRK2_{wv} and mGIRK2_{wt}. Each line displays the position of one ion on the z-axis over simulation time. The positions of the backbone oxygens of the SF residues, which frame the canonical ion binding sites S0-S4, are shown in black. c Presence of water in the mGIRK2_{wv} and mGIRK2_{wt}. Each line displays the position of one water molecule on the z-axis over simulation time. The positions of the backbone oxygens of the SF residues are shown in black.” Lines 122-124: “Figure 2b shows the outward permeation of ions and the presence of water in the mGIRK2_{wv} SF and mGIRK2_{wt} SF over simulation time. The most striking divergence between the simulations is the mutation-induced loss of selectivity for K⁺, which is demonstrated by several Na⁺ passing through the mGIRK2_{wv} SF.” Lines 138-144: “In the simulations with mGIRK2_{wv}, it can be observed that raised Na⁺ permeation levels are accompanied by a deficient exclusion of water from the SF. The right permeation plot in Figure 2b shows water entering the mutant SF after about 250 ns simulation time, occurring simultaneously to the first Na⁺ permeation event. However, solvation is not tightly coupled to Na⁺ permeation, since water is also present in case only K⁺ occupies the SF (e.g. Fig. 2b between 1.5 and 2 μs). On the contrary and in line with previous studies ^{7,21,24}, no water can be seen in the simulations with mGIRK2_{wt}. K⁺ ions shed the water shell upon entering the SF and permeate in a dehydrated manner.”
5) Lines 159-161, please delete reference to Fig. 3e, as it does not show Ba²⁺ block. Sentence ought to read as follows: "I-V curves of GIRK2G154S with or without Gβγ showed little rectification and were	We apologize for the mistake; the text has been altered accordingly. Reference to Fig. 4f has been added, which demonstrates the little rectification of hGIRK2_{G154S} with or without Gβγ. Furthermore, we have clarified that a partial block by Ba²⁺ still occurred in the hGIRK2_{G152S} channel, when	Lines 186-188: “I-V curves of hGIRK2_{G154S} with or without Gβγ showed little rectification (Fig. 4f) and reduced block by Ba²⁺ (~60% block at 1 mM Ba²⁺, compared to full block of hGIRK2_{wt}) (Supplementary Fig. 2g).” Supplementary Fig. 2: listed at the end of this letter

not blocked by Ba²⁺ (Supplementary Fig. 2f)".	currents were measured at -80 mV: "...and reduced block by Ba²⁺ (~60% block at 1 mM Ba²⁺, compared to full block of hGIRK2_{wt})". This is shown in new Supplementary Fig. 2g.	
6) Lines 294-295, I believe the authors meant to cite Fig. 7 not Fig. 8. The sentence ought to read: "Functional Mode Analysis (FMA) unveiled a correlation of aberrant SF dynamics at residue I155 with movements of residues previously reported to contact G$\beta\gamma$6 (Fig. 7)".	We again apologize for the mistake, the figure reference was changed as remarked.	Line 392-393: "More specifically, Functional Mode Analysis (FMA) unveiled a correlation of aberrant SF dynamics at residue I155 with movements of residues previously reported to contact G$\beta\gamma$6 (Fig. 7)."
7) Lines 332-333, I would replace the word "alleviate" with "prevent" so that the sentence reads: "Considering the substantial energy barrier associated with Na⁺ dehydration, the insufficient exclusion of H₂O is likely to prevent the passing of Na⁺ flux through GIRK2_{WT}".	We thank reviewer 1 for the suggestion. We included "prevent" in the sentence and changed the wording to emphasize our point.	Lines 458-460: "Considering the substantial energy barrier associated with Na⁺ dehydration, the exclusion of water is likely to prevent the passing of Na⁺ flux through mGIRK2_{wt}."
8) Lines 477-480: "The runs were carried out under different electric fields of 40 mV nm⁻¹, 50 mV nm⁻¹ and 60 mV nm⁻¹ along the z-axis. Taking into account a z-axis box length of ~14.5 nm, the electric fields yield transmembrane potentials of ~580 mV, ~725 mV and 870 mV, respectively." I believe this is not correct. The thickness of the membrane should be used	We thank reviewer 1 for this comment and calculation, we checked and corrected the text as suggested.	Lines 602-604: "Taking into account the thickness of the membrane of ~3.5 nm, the electric fields yield transmembrane potentials of 140 mV, 175 mV and 210 mV, respectively."

(about 3.5nm) rather than the length of the simulation box. The transmembrane potentials should thus be about 140mV, 175mV, and 210mV for the simulations.		
9) Dr. Dascal should complete the "Acknowledgements" and "Author Contributions" sections.	We apologize for this error; the respective segments were completed.	Lines 785-804: * *please look at the corresponding lines in the revised manuscript
10) I would indicate either mGIRK2 for mouse or hGIRK2 for human to always be clear which channel is being used.	The channels were renamed as suggested. The sentence explaining the nomenclature was altered. The names of the channels were altered throughout the manuscript (including figure captions).	We introduced the abbreviations consistently throughout the manuscript.

Reviewer 2

Point raised	Response	Changes in manuscript
Major point		
The major novel findings in this manuscript are the allosteric effects in the CTD caused by the weaver mutation. The authors state that "simulations with Gβγ bound GIRK2, which go beyond the scope of this study due to the inherent computational effort, might be necessary to make traces of a coupling between the SF and Gβγ binding site visible" which is reasonable. However, absent such simulations there is not sufficient experimental evidence to support their MD findings in the manuscript as presented. The authors use the decrease in barium affinity of the wild-type Gβγ bound channel as evidence to support the allosteric	We thank reviewer 2 for the valuable input. The suggestion to examine changes in PIP ₂ or alcohol affinities in the hGIRK2 _{G152S} mutant is based on the interpretation that the mutation changes the affinity of GIRK2 to Gβγ. However, our simulations do not predict such a change, but only reveal an allosteric link between the dynamics of the SF and the amino acids close to, or participating in, Gβγ binding. The mutation, and the resulting increased mobility of I155 and the new H-bonds behind the SF, may affect this allosteric link in a way that the channel becomes constitutively active. Therefore, we cannot make predictions whether the Gβγ affinity is changed at all in the mutant channel. Similar argumentation applies to the effect of the mutation of alcohol binding. Thus, although we find the suggestions very interesting and tantalizing, we do not think that they will be mechanistically interpretable without an intensive MD	Lines 419-425: "MD simulations with mGIRK2 _{wv} show that the mutation G156S leads to aberrant movements and an altered geometry of the SF (Fig. 5b, c; Fig. 6), which are likely to affect inherent features of the ion channel, such as selectivity and conductivity. Moreover, the mutation might alter the coupling between the SF and Gβγ binding site, thereby providing a possible explanation for the impaired mGIRK2 _{wv} activation by Gβγ observed in experiments ^{14,17,19} . However, it is unclear whether the mutation changes the affinity of modulators binding to the Gβγ-binding site, such as ethanol or the Gβγ subunit. This aspect remains to be elucidated in future studies, since the required modeling is beyond the scope of this framework."

coupling of the SF and the Gβγ binding site. Although this is one explanation for these data, it is not the only possible explanation and is not sufficient to support the MD findings. An additional experiment that would support the MD simulation finding would strengthen this result. Because Gβγ binding affects PIP2 affinity, a decrease in Gβγ binding would be expected to cause a concomitant decrease in PIP2 binding, so the authors could measure PIP2 affinity to support their hypothesis. Alternatively, alcohol binds GIRK2 channels in the same region of the CTD dynamic changes were described in this study. Therefore, one would expect a decrease in alcohol affinity and, in turn, G-protein independent activation by alcohol in the mutant channel. Either of these experiments would provide additional experimental support that would significantly strengthen the findings of their MD simulations.	modeling effort, which is out of scope of this paper. In order to reflect on this idea, we included a short paragraph in the discussion about the putative effect of the SF mutation on modulator affinities to or near the Gβγ binding site. We further agree that the Ba²⁺ block experiment provides only initial and partial experimental support to the link between Gβγ binding and the SF. Nevertheless, our conclusions here are also supported by the MD simulations of Li et al. (ref. 42), as indicated in the Discussion: “Recently, Li et al.⁴² described the GIRK2 SF as a crucial determinant for K⁺ permeation, which can be influenced by binding of Gβγ and Na⁺”. We agree that future experiments, probably driven by further MD modeling, will be needed to further scrutinize the hypothesis of allosteric regulation of the SF by Gβγ binding.	
Minor points		
1) The authors make much of the fact that this paper presents the first biophysical characterization of the human GIRK2 G154S mutation and present this as a major advancement in the field. The human and mouse GIRK2 protein sequences are almost	We thank reviewer 2 for this important remark and agree on this point. We altered the manuscript in order to take the emphasis off the biophysical characterization of hGIRK2_{G154S}, and describe it more as confirmatory.	Line 32: “We further present a first-time functional characterization.” Line 80-82: “In this work, we use μs long MD simulations to investigate the effect of the weaver mutation on mGIRK2, which we corroborate by a functional characterization of the corresponding human mutant, hGIRK2_{G154S}, with electrophysiology experiments.”

identical (depending on the isoform) and there are no differences between these species in the biophysical properties characterized in this study. Furthermore, the MD simulations use structures solved using the mouse GIRK channel, so the import of the findings in this study are predicated on the mechanism being conserved between species. The manuscript would be improved if the human biophysical characterization was described more as confirmatory (which it is), rather than a significant advancement of the field.		Line 453-456: “We here present the first time functional and structural characterization of the human GIRK2_{G154S} disease mutant, which we show to exhibit properties strongly reminiscent of mGIRK2_{ww}, including loss of inward rectification, acquisition of sensitivity to block by the sodium channel blocker QX314, high constitutive activity and reduced K⁺ selectivity^{14,16,19}.”
2. Although the new GIRK2 cryo-EM structures are mentioned in the introduction, there is no mention of how these new structures, and the novel extended conformation exhibited in the absence of PIP₂, might affect the interpretation of how the selectivity filter might couple to the Gβγ binding region. The manuscript would be improved by at least a mention of such an interpretation.	We have now included the interpretation of the coupling between the SF and the Gβγ binding site with regard to the mechanism observed in the cryo-EM structures.	Lines 439-449: “MD simulations performed in this study do not take into account the role of the CTD conformation in GIRK2 gating, as it was recently suggested by Cryo-EM structures¹⁰. More precisely, the CTD was reported to switch from an extended to a docked conformation upon PIP₂ binding, which makes GIRK2 susceptible to Gβγ binding. Since modelling of the different CTD conformations is beyond the scope of this study, it remains to be elucidated how the coupling between the SF and the Gβγ binding site is influenced by conformational changes of the CTD, or whether conformational changes of the CTD themselves affect the SF. However, the MD simulations in this framework were carried out with the PIP₂-bound mGIRK2 channel, which, according to the mechanism proposed by the Cryo-EM structures, adopts the docked conformation. Under these conditions, we do not expect a conformational change of the CTD. Thus, the coupling between the SF and the Gβγ binding site may be considered valid for the PIP₂-bound channel, independent of conformational changes upon PIP₂ removal.”

Updated figures:

Figure 1:

A high resolution figure was introduced in order to adhere to the figure formatting guidelines.

Fig 1. The structure of the GIRK2 channel and its regulators. Two of the four subunits of the channel as well as channel modulators are illustrated.

Figure 2:

The figure labels (a), b), c)) were introduced in order to adhere to the figure formatting guidelines. The labels from GIRK2_{wv} and GIRK2_{wt} were changed to mGIRK2_{wv} and mGIRK2_{wt}. Furthermore, the figure caption was changed (motivated by Minor Point 4 of Reviewer 1).

Fig. 2 Ion and water flux through the mGIRK2_{wv} (top) and mGIRK2_{wt} (bottom) SF over simulation time. **a** Representative snapshots of ion and water flux through the SF. mGIRK2_{wv} is colored blue, while mGIRK2_{wt} is colored green. The SF is depicted in cartoon representation with the backbone oxygens of the SF residues indicated as red sticks. K⁺ ions are represented as pink spheres, while Na⁺ ions are represented as smaller, light blue spheres. Water molecules are shown in a ball-and-stick representation. **b** K⁺ (purple) and Na⁺ ions (blue) flux observed in runs with mGIRK2_{wv} and mGIRK2_{wt}. Each line displays the position of one ion on the z-axis over simulation time. The positions of the backbone oxygens of the SF residues, which frame the canonical ion binding sites S0-S4, are shown in black. **c** Presence of water in the mGIRK2_{wv} and mGIRK2_{wt}. Each line displays the position of one water molecule on the z-axis over simulation time. The positions of the backbone oxygens of the SF residues are shown in black.

Figure 5 b, c:

Minor stylistic changes were made in the formatting of the y-axis of the two plots in b) and c). In the previous version, the plots were nearer to each other and shared the y-axis labels. :

Fig. 5 Differences in ion occupancies and in the hydrogen bond network between mGIRK2_{wt} and mGIRK2_{wv}. **a** Top: Ion occupancies in the mGIRK2_{wv} SF observed over a total of 9 μs long MD simulation. Two characteristic snapshots of Na⁺ and K⁺ permeation are shown on the right. Bottom: Ion occupancies in the wild-type SF observed over 6 μs long MD simulations. The snapshots on the right show a characteristic K⁺ occupancy pattern as well as Na⁺ block between site S3 and S4. **b** Top: Representative snapshot of hydrogen bonds between the sidechain of S156 and the backbone oxygen of I155 in the mGIRK2_{wv} SF. Bottom: Density distribution of the distances and the numbers of hydrogen bonds between the sidechain of S156 and the backbone oxygen of I155 observed over 9 μs simulations with GIRK2_{wv}. **c** Top: Representative snapshot of hydrogen bonds between the sidechain of S156 and a water behind the mGIRK2_{wv} SF. Bottom: Histogram of the number of water molecules behind the SF, which form hydrogen bonds to the sidechain of S156, over 9 μs simulations with mGIRK2_{wv}.

Figure 6:

Figure labels were introduced in order to adhere to the figure formatting guidelines. The y-axis label of Figure 6 a) was corrected. Furthermore, the plot displaying the SF conformations, which show the minimum and maximum distance between opposing I155 C α atoms, was altered in a way that the conformations are displayed in two different figures (figure c and figure d).

Fig. 6 Structural aberrations in the SF observed over 9 μ s mGIRK2_{wv} and 6 μ s mGIRK2_{wt} simulation. a Distributions of distances between C α of opposing mGIRK2 SF residues. **b** Phi-angle distributions of mGIRK2 SF residues. **c and d** mGIRK2_{wv} SF conformations showing the minimum (figure c)

and maximum (figure d) distance between opposing I155 Ca atoms. The Ca atoms of I155 is highlighted are shown in dark red. Other Ca atoms of the SF residues are shown as salmon spheres.

Supplementary Fig. 2:

Subplots 2 e) and f) were changed in order to reflect on a partial block by Ba^{2+} , which still occurred in the hGIRK2_{G152S} channel, when currents were measured at -80 mV. (Minor point 5, reviewer 1)

Supplementary Figure 2. The effect of Ba²⁺ and QX314 on naïve as well as hGIRK2_{wt} and hGIRK2_{G154S}-expressing oocytes. **a** Average currents from naïve oocytes obtained in four external K⁺ concentrations (n=5). **b-d** Exemplary I-V relationships of hGIRK2_{wt} obtained in four external K⁺ concentrations. I-V curves obtained in the presence of 1 mM Ba²⁺ (c) were subtracted from I-V curves without the blocker (b), yielding net GIRK I-V

curves (d). **e** Exemplary I-V relationships of hGIRK2_{G154S} obtained in four external K⁺ concentrations with and without 400 μM QX314. **f** I-V curves of QX sensitive currents in four external K⁺ concentrations. **g** Normalized data of % inhibition by 1 mM Ba on hGIRK2_{wt} vs hGIRK2_{G154S} with and without Gβγ. (N=2). Number of oocytes are shown within the bars. One way ANOVA followed by Kruskal-Wallis test. ***p<0.001

Supplementary Fig. 3:

This Figure was newly introduced and shows the results obtained for the HBC gate analysis (Major point, reviewer 1).

Supplementary Figure 3. Pore dimensions at the HBC gate over 6 μs mGIRK2_{wt} and 9 μs mGIRK2_{wv} MD simulation. a Distribution of the distances between the Ca of opposing F192. **b** Distribution of minimum distances between opposing F192.

REVIEWERS' COMMENTS:

Reviewer #1 (Remarks to the Author):

The authors carefully considered each and everyone of my critical comments and have addressed them satisfactorily. I do not have any further critical remarks for this very fine study, thus I endorse it fully.

Reviewer #2 (Remarks to the Author):

I believe that the changes in the response to the reviews of the manuscript "A disease mutation provides insights into gating regulation of a K⁺ channel" by Friesacher et al. adequately address the concerns and make the manuscript suitable for publication in Communications Biology.